# Consistency Diffusion Bridge Models

**Guande He**[†1*]   **Kaiwen Zheng**[†1*]   **Jianfei Chen**[1],   **Fan Bao**[12],   **Jun Zhu**[‡123]

[1]Dept. of Comp. Sci. & Tech., Institute for AI, BNRist Center, THBI Lab
[1]Tsinghua-Bosch Joint ML Center, Tsinghua University, Beijing, China
[2]Shengshu Technology, Beijing    [3]Pazhou Lab (Huangpu), Guangzhou, China
guande.he17@outlook.com; zkwthu@gmail.com;
fan.bao@shengshu.ai; {jianfeic, dcszj}@tsinghua.edu.cn

## Abstract

Diffusion models (DMs) have become the dominant paradigm of generative modeling in a variety of domains by learning stochastic processes from noise to data. Recently, diffusion denoising bridge models (DDBMs), a new formulation of generative modeling that builds stochastic processes between fixed data endpoints based on a reference diffusion process, have achieved empirical success across tasks with coupled data distribution, such as image-to-image translation. However, DDBM's sampling process typically requires hundreds of network evaluations to achieve decent performance, which may impede their practical deployment due to high computational demands. In this work, inspired by the recent advance of consistency models in DMs, we tackle this problem by learning the consistency function of the probability-flow ordinary differential equation (PF-ODE) of DDBMs, which directly predicts the solution at a starting step given any point on the ODE trajectory. Based on a dedicated general-form ODE solver, we propose two paradigms: consistency bridge distillation and consistency bridge training, which is flexible to apply on DDBMs with broad design choices. Experimental results show that our proposed method could sample $4\times$ to $50\times$ faster than the base DDBM and produce better visual quality given the same step in various tasks with pixel resolution ranging from $64 \times 64$ to $256 \times 256$, as well as supporting downstream tasks such as semantic interpolation in the data space.

## 1   Introduction

Diffusion models (DMs) [53, 21, 60] have reached unprecedented levels as a family of generative models in various areas, including image generation [10, 50, 48], audio synthesis [5, 45], video generation [20], as well as image editing [41, 42], solving inverse problems [25, 56], and density estimation [59, 28, 37, 71]. In the era of AI-generated content, the stable training, scalability & state-of-the-art generation performance of DMs successfully make them serve as the fundamental component of large-scale, high-performance text-to-image [14] and text-to-video [18, 2] models.

A critical characteristic of diffusion models is their iterative sampling procedure, which progressively drives random noise into the data space. Although this paradigm yields a sample quality that stands out from other generation models, such as VAEs [29, 46], GANs [17], and Normalizing Flows [11, 12, 30], it also results in a notoriously lower sampling efficiency compared to other arts. In response to this, consistency models [58] have emerged as an attractive family of generative models by learning a consistency function that directly predicts the solution of a probability-flow ordinary differential equation (PF-ODE) at a certain starting timestep given any points in the ODE trajectory, designed to be a one-step generator that directly maps noise to data. Consistency models can be

---

*Work done during an internship at Shengshu;    †Equal contribution;    ‡The corresponding author.

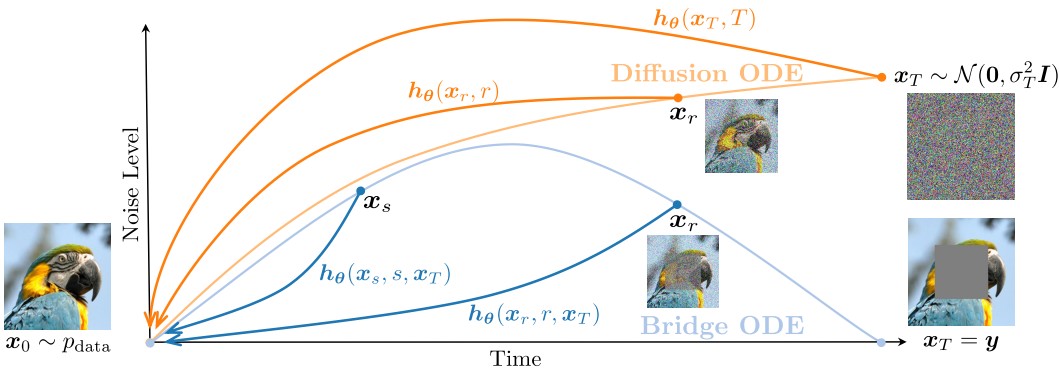

Figure 1: Illustration of consistency models (CMs) on PF-ODEs of diffusion models and our proposed consistency diffusion bridge models (CDBMs) building on PF-ODEs of diffusion bridges. Different from diffusion models, the PF-ODE of diffusion bridge is only well defined in $t < T$ due to the singularity induced by the fixed terminal endpoint. To this end, a valid input for CDBMs is some $\boldsymbol{x}_t$ for $t < T$, which is typically obtained by one-step posterior sampling with a coarse estimation of $\boldsymbol{x}_0$ with an initial network evaluation.

naturally integrated with diffusion models by adapting the score estimator of DMs to a consistency function of their PF-ODE via distillation [58, 26] or fine-tuning [15], showing promising performance for few-step generation in various applications like latent space [40] and video [64].

Despite the remarkable achievements in generation quality and better sampling efficiency, a fundamental limitation of diffusion models is that their prior distribution is usually restricted to a non-informative Gaussian noise, due to the nature of their underlying data to noise stochastic process. This characteristic may not always be desirable when adopting diffusion models in some scenarios with an informative non-Gaussian prior, such as image-to-image translation. Alternatively, an emergent family of generative models focuses on leveraging diffusion bridges, a series of altered diffusion processes conditioned on given endpoints, to model transport between two arbitrary distributions [44, 36, 33, 54, 51, 72, 7]. Among them, denoising diffusion bridge models (DDBMs) [72] study the reverse-time diffusion bridge conditioned on the terminal endpoint, and employ simulation-free, non-iterative training techniques for it, showing superior performance in application with coupled data pairs such as distribution translation compared to diffusion models. However, DDBMs generally require hundreds of network evaluations to produce samples with decent quality, even using an advanced high-order hybrid sampler, potentially hindering their deployments in real-world applications.

In this work, inspired by recent advances in consistency models with diffusion ODEs [58, 57, 15], we introduce consistency diffusion bridge models (CDBMs) and develop systematical techniques to learn the consistency function of the PF-ODEs in DDBMs for improved sampling efficiency. Firstly, to facilitate flexible integration of consistency models in DDBMs, we present a unified perspective on their design spaces, including noise schedule, prediction target, and network parameterizations, termed the same as in diffusion models [28, 24]. Additionally, we derive a first-order ODE solver based on the general-form noise schedule. This universal framework largely decouples the formulation of DDBMs and the corresponding consistency models from highly practical design spaces, allowing us to reuse the successful empirical choices of various diffusion bridges for CDBMs regardless of their different theoretical premises. On top of this, we then propose two paradigms for training CDBMs: consistency bridge distillation and consistency bridge training. This approach is free of dependence on a restricted form of noise schedule and the corresponding Euler ODE solver as in previous work [58], thus enhancing the practical versatility and extensibility of the CDBM framework.

We verify the effectiveness of CDBMs in two applications: image translation and image inpainting by distilling or fine-tuning DDBMs with various design spaces. Experimental results demonstrate that our approach can improve the sampling speed of DDBMs from $4\times$ to $50\times$, in terms of the Fréchet inception distance [19] (FID) evaluated with two-step generation. Meanwhile, given the same computational budget, CDBMs have better performance trade-offs compared to DDBMs, both quantitatively and qualitatively. CDBMs also retain the desirable properties of generative modeling, such as sample diversity and the ability to perform semantic interpolation in the data space.

## 2 Preliminaries

### 2.1 Diffusion Models

Given the data distribution $p_{\text{data}}(\boldsymbol{x}), \boldsymbol{x} \in \mathbb{R}^m$, diffusion models [53, 21, 60] specify a forward-time diffusion process from an initial data distribution $p_0 = p_{\text{data}}$ to a terminal distribution $p_T$ within a finite time horizon $t \in [0, T]$, defined by a stochastic differential equation (SDE):

$$\mathrm{d}\boldsymbol{x}_t = \boldsymbol{f}(\boldsymbol{x}_t, t)\mathrm{d}t + g(t)\mathrm{d}\boldsymbol{w}_t, \quad \boldsymbol{x}_0 \sim p_0, \tag{1}$$

where $\boldsymbol{w}_t$ is a standard Wiener process, $\boldsymbol{f} : \mathbb{R}^m \times [0, T] \to \mathbb{R}^m$ and $g : [0, T] \to \mathbb{R}^d$ are drift and diffusion coefficients, respectively. The terminal distribution $p_T$ is usually designed to approximate a tractable prior $p_{\text{prior}}$ (e.g., standard Gaussian) with the appropriate choice of $\boldsymbol{f}$ and $g$. The corresponding reverse SDE and the probability flow ordinary differential equation (PF-ODE) of the forward SDE in Eqn. (1) is given by [1, 60]:

$$\mathrm{d}\boldsymbol{x}_t = [\boldsymbol{f}(\boldsymbol{x}_t, t) - g^2(t)\nabla \log p_t(\boldsymbol{x}_t)]\mathrm{d}t + g(t)\mathrm{d}\bar{\boldsymbol{w}}_t, \quad \boldsymbol{x}_T \sim p_T \approx p_{\text{prior}}, \tag{2}$$

$$\mathrm{d}\boldsymbol{x}_t = \left[\boldsymbol{f}(\boldsymbol{x}_t, t) - \frac{1}{2}g^2(t)\nabla \log p_t(\boldsymbol{x}_t)\right]\mathrm{d}t, \quad \boldsymbol{x}_T \sim p_T \approx p_{\text{prior}}, \tag{3}$$

where $\bar{\boldsymbol{w}}_t$ is a reverse-time standard Wiener process and $p_t(\boldsymbol{x}_t)$ is the marginal distribution of $\boldsymbol{x}_t$. Both the reverse SDE and PF-ODE can act as a generative model by sampling $\boldsymbol{x}_T \sim p_{\text{prior}}$ and simulating the trajectory from $\boldsymbol{x}_T$ to $\boldsymbol{x}_0$. The major difficulty here is that the score function $\nabla \log p_t(\boldsymbol{x}_t)$ remains unknown, which can be approximated by a neural network $s_{\boldsymbol{\theta}}(\boldsymbol{x}_t, t)$ with *denoising score matching* [63]:

$$\mathbb{E}_{t \in \mathcal{U}(0,T)}\mathbb{E}_{p_0(\boldsymbol{x}_0)p_{t|0}(\boldsymbol{x}_t|\boldsymbol{x}_0)}\left[\lambda(t)\|s_{\boldsymbol{\theta}}(\boldsymbol{x}_t, t) - \nabla \log p_{t|0}(\boldsymbol{x}_t|\boldsymbol{x}_0)\|_2^2\right], \tag{4}$$

where $\mathcal{U}(0, T)$ is uniform distribution, $\lambda(t) > 0$ is a weighting function, and $p_{t|0}(\boldsymbol{x}_t|\boldsymbol{x}_0)$ is the transition kernel from $\boldsymbol{x}_0$ to $\boldsymbol{x}_t$. A common practice is to use a linear drift $f(t)\boldsymbol{x}_t$ such that $p_{t|0}(\boldsymbol{x}_t|\boldsymbol{x}_0)$ is an analytic Gaussian distribution $\mathcal{N}(\alpha_t\boldsymbol{x}_0, \sigma_t^2\boldsymbol{I})$, where $\alpha_t = e^{\int_0^t f(\tau)\mathrm{d}\tau}, \sigma_t^2 = \alpha_t^2\int_0^t \frac{g^2(\tau)}{\alpha_\tau^2}\mathrm{d}\tau$ is defined as the *noise schedule* [28]. The resulting score predictor $s_{\boldsymbol{\theta}}(\boldsymbol{x}_t, t)$ can replace the true score function in Eqn. (2) and (3) to obtain the empirical diffusion SDE and ODE, which can be simulated by various SDE or ODE solvers [55, 38, 39, 16, 70].

### 2.2 Consistency Models

Given a trajectory $\{\boldsymbol{x}_t\}_{t=\epsilon}^T$ with a fixed starting timestep $\epsilon$ of a PF-ODE, consistency models [58] aim to learn the solution of the PF-ODE at $t = \epsilon$, also known as the *consistency function*, defined as $\boldsymbol{h} : (\boldsymbol{x}_t, t) \mapsto \boldsymbol{x}_\epsilon$. The optimization process for consistency models contains the online network $\boldsymbol{h}_{\boldsymbol{\theta}}$ and a reference target network $\boldsymbol{h}_{\boldsymbol{\theta}^-}$, where $\boldsymbol{\theta}^-$ refers to $\boldsymbol{\theta}$ with operation stopgrad, i.e., $\boldsymbol{\theta}^- = \text{stopgrad}(\boldsymbol{\theta})$. The networks are hand-designed to satisfy the boundary condition $\boldsymbol{h}_{\boldsymbol{\theta}}(\boldsymbol{x}_\epsilon, \epsilon) = \boldsymbol{x}_\epsilon$, which can be typically achieved with proper parameterization on the neural network. For PF-ODE taking the form in Eqn. (3) with a linear drift $f(t)\boldsymbol{x}_t$, the overall learning objective of consistency models can be described as:

$$\mathbb{E}_{t \in \mathcal{U}(\epsilon,T), r=r(t)}\mathbb{E}_{p_0(\boldsymbol{x}_0)p_{t|0}(\boldsymbol{x}_t|\boldsymbol{x}_0)}\left[\lambda(t)d\left(\boldsymbol{h}_{\boldsymbol{\theta}}(\boldsymbol{x}_t, t), \boldsymbol{h}_{\boldsymbol{\theta}^-}(\hat{\boldsymbol{x}}_r, r)\right)\right], \tag{5}$$

where $r(t)$ is a function that specifies another timestep $r$ (usually with $t > r$), $d$ denotes some metric function with $\forall \boldsymbol{x}, \boldsymbol{y} : d(\boldsymbol{x}, \boldsymbol{y}) \geq 0$ and $d(\boldsymbol{x}, \boldsymbol{y}) = 0$ iff. $\boldsymbol{x} = \boldsymbol{y}$. Here $\hat{\boldsymbol{x}}_r$ is a function that estimates $\boldsymbol{x}_r = \boldsymbol{x}_t + \int_t^r \frac{\mathrm{d}\boldsymbol{x}_\tau}{\mathrm{d}\tau}\mathrm{d}\tau$, which can be done by simulating the empirical diffusion ODE with a pre-trained score predictor $s_{\boldsymbol{\phi}}(\boldsymbol{x}_t, t)$ or empirical score estimator $-\frac{\boldsymbol{x}_t - \alpha_t\boldsymbol{x}_0}{\sigma_t^2}$. The corresponding learning paradigms are named *consistency distillation* and *consistency training*, respectively.

### 2.3 Denoising Diffusion Bridge Models

Given a data pair sampled from an arbitrary unknown joint distribution $(\boldsymbol{x}, \boldsymbol{y}) \sim q_{\text{data}}(\boldsymbol{x}, \boldsymbol{y}), \boldsymbol{x}, \boldsymbol{y} \in \mathbb{R}^m$ and let $\boldsymbol{x}_0 = \boldsymbol{x}$, *denoising diffusion bridge models* (DDBMs) [72] specify a stochastic process

that ensures $\boldsymbol{x}_T = \boldsymbol{y}$ almost surly via applying *Doob's h-transform* [13, 47] on a reference diffusion process in Eqn. (1):

$$\mathrm{d}\boldsymbol{x}_t = \left[\boldsymbol{f}(\boldsymbol{x}_t, t) + g^2(t)\nabla_{\boldsymbol{x}_t} \log p_{T|t}(\boldsymbol{x}_T = \boldsymbol{y}|\boldsymbol{x}_t)\right]\mathrm{d}t + g(t)\mathrm{d}\boldsymbol{w}_t, \quad (\boldsymbol{x}_0, \boldsymbol{x}_T) = (\boldsymbol{x}, \boldsymbol{y}) \sim q_{\mathrm{data}}, \tag{6}$$

where $p_{T|t}(\boldsymbol{x}_T = \boldsymbol{y}|\boldsymbol{x}_t)$ is the transition kernel of the reference diffusion process from $t$ to $T$, evaluated at $\boldsymbol{x}_T = \boldsymbol{y}$. Denoting the marginal distribution of Eqn. (6) as $\{q_t\}_{t=0}^T$, it can be shown that the forward bridge SDE in Eqn. (6) is characterized by the diffusion distribution conditioned on both endpoints, that is, $q_{t|0T}(\boldsymbol{x}_t|\boldsymbol{x}_0, \boldsymbol{x}_T) = p_{t|0T}(\boldsymbol{x}_t|\boldsymbol{x}_0, \boldsymbol{x}_T)$, which is an analytic Gaussian distribution. A generative model can be obtained by modeling $q_{t|T}(\boldsymbol{x}_t|\boldsymbol{x}_T = \boldsymbol{y})$, whose reverse SDE and PF-ODE are given by:

$$\mathrm{d}\boldsymbol{x}_t = \left[\boldsymbol{f}(\boldsymbol{x}_t, t) - g^2(t)\left(\nabla_{\boldsymbol{x}_t}\log q_{t|T}(\boldsymbol{x}_t|\boldsymbol{x}_T = \boldsymbol{y}) - \nabla_{\boldsymbol{x}_t}\log p_{T|t}(\boldsymbol{x}_T = \boldsymbol{y}|\boldsymbol{x}_t))\right)\right]\mathrm{d}t + g(t)\mathrm{d}\bar{\boldsymbol{w}}_t, \tag{7}$$

$$\mathrm{d}\boldsymbol{x}_t = \left[\boldsymbol{f}(\boldsymbol{x}_t, t) - g^2(t)\left[\frac{1}{2}\nabla_{\boldsymbol{x}_t}\log q_{t|T}(\boldsymbol{x}_t|\boldsymbol{x}_T = \boldsymbol{y}) - \nabla_{\boldsymbol{x}_t}\log p_{T|t}(\boldsymbol{x}_T = \boldsymbol{y}|\boldsymbol{x}_t))\right]\right]\mathrm{d}t. \tag{8}$$

The only unknown term remains is the score function $\nabla_{\boldsymbol{x}_t}\log q_{t|T}(\boldsymbol{x}_t|\boldsymbol{x}_T = \boldsymbol{y})$, which can be estimated with a neural network $s_{\boldsymbol{\theta}}(\boldsymbol{x}_t, t, \boldsymbol{y})$ via *denoising bridge score matching* (DBSM):

$$\mathbb{E}_{t \in \mathcal{U}(0,T)}\mathbb{E}_{q_{\mathrm{data}}(\boldsymbol{x}, \boldsymbol{y})q_{t|0T}(\boldsymbol{x}_t|\boldsymbol{x}_0 = \boldsymbol{x}, \boldsymbol{x}_T = \boldsymbol{y})}\left[\lambda(t)\|s_{\boldsymbol{\theta}}(\boldsymbol{x}_t, t, \boldsymbol{y}) - \nabla\log q_{t|0T}(\boldsymbol{x}_t|\boldsymbol{x}_0 = \boldsymbol{x}, \boldsymbol{x}_T = \boldsymbol{y})\|_2^2\right]. \tag{9}$$

Replacing $\nabla_{\boldsymbol{x}_t}\log q_{t|T}(\boldsymbol{x}_t|\boldsymbol{x}_T = \boldsymbol{y})$ in Eqn. (7) and (8) with the learned score predictor $s_{\boldsymbol{\theta}}(\boldsymbol{x}_t, t, \boldsymbol{y})$ would yield the empirical bridge SDE and ODE that could be solved for generation purposes.

## 3 Consistency Diffusion Bridge Models

In this section, we introduce consistency diffusion bridge models, extending the techniques of consistency models to DDBMs to further boost their performance and sample efficiency. Define the consistency function of the bridge ODE in Eqn. (8) as $\boldsymbol{h} : (\boldsymbol{x}_t, t, \boldsymbol{y}) \mapsto \boldsymbol{x}_\epsilon$ with a given starting timestep $\epsilon$, our goal is to learn the consistency function using a neural network $\boldsymbol{h}_{\boldsymbol{\theta}}(\cdot, \cdot, \boldsymbol{y})$ with the following high-level objective similar to Eqn. (5):

$$\mathbb{E}_{t \in \mathcal{U}(\epsilon, T), r = r(t)}\mathbb{E}_{q_{\mathrm{data}}(\boldsymbol{x}, \boldsymbol{y})q_{t|0T}(\boldsymbol{x}_t|\boldsymbol{x}_0 = \boldsymbol{x}, \boldsymbol{x}_T = \boldsymbol{y})}\left[\lambda(t)d\left(\boldsymbol{h}_{\boldsymbol{\theta}}(\boldsymbol{x}_t, t, \boldsymbol{y}), \boldsymbol{h}_{\boldsymbol{\theta}^-}(\hat{\boldsymbol{x}}_r, r, \boldsymbol{y})\right)\right]. \tag{10}$$

To begin with, we first present a unified view of the design spaces such as noise schedule, network parameterization & precondition, as well as a general ODE solver for DDBMs. This allows us to: (1) decouple the successful practical designs of previous diffusion bridges from their different theoretical premises; (2) decouple the framework of consistency models from certain design choices of the corresponding PF-ODE, such as the reliance on VE schedule with Euler ODE solver of the original derivation of consistency models [58]. This would largely facilitate the development of consistency models that utilize the rich design spaces of existing diffusion bridges on DDBMs in a universal way. Then, we elaborate on two ways to train $\boldsymbol{h}_{\boldsymbol{\theta}}$ based on different choices of $\hat{\boldsymbol{x}}_r$, consistency bridge distillation, and consistency bridge training, with the proposed unified design spaces.

### 3.1 A Unified View on Design Spaces of DDBMs

**Noise Schedule** We consider the linear drift $f(t)\boldsymbol{x}_t$ and define:

$$\alpha_t = e^{\int_0^t f(\tau)\mathrm{d}\tau}, \quad \bar{\alpha}_t = e^{-\int_t^T f(\tau)\mathrm{d}\tau}, \quad \rho_t^2 = \int_0^t \frac{g^2(\tau)}{\alpha_\tau^2}\mathrm{d}\tau, \quad \bar{\rho}_t^2 = \int_t^T \frac{g^2(\tau)}{\alpha_\tau^2}\mathrm{d}\tau, \tag{11}$$

which aligns with the common notation of noise schedules used in diffusion models by denoting $\sigma_t = \alpha_t\rho_t$. Then we could express the analytic conditional distributions of DDBMs as follows:

$$q_{t|0T}(\boldsymbol{x}_t|\boldsymbol{x}_0, \boldsymbol{x}_T) = p_{t|0T}(\boldsymbol{x}_t|\boldsymbol{x}_0, \boldsymbol{x}_T) = \mathcal{N}\left(a_t\boldsymbol{x}_T + b_t\boldsymbol{x}_0, c_t^2\boldsymbol{I}\right),$$

$$\text{where} \quad a_t = \frac{\bar{\alpha}_t\rho_t^2}{\rho_T^2}, \quad b_t = \frac{\alpha_t\bar{\rho}_t^2}{\rho_T^2}, \quad c_t^2 = \frac{\alpha_t^2\bar{\rho}_t^2\rho_t^2}{\rho_T^2}. \tag{12}$$

The form of $q_{t|0T}$ is consistent with the original formulation of DDBM in [72]. Here, inspired by [6], we opt to adopt a more neat set of notations for enhanced compatibility. As shown in Table 1, with such notations, we could easily unify the design choices for diffusion bridges [33, 72, 6] that have shown effectiveness in various tasks and expeditiously employ consistency models on top of them.

Table 1: Specifications of design spaces in different diffusion bridges. The details of network parameterization are in Appendix B.4 due to space limit.

| | Brownian Bridge | I2SB [33] | DDBM [72] | | Bridge-TTS [6] | |
| | *default* | *default*† | *VP*‡ | *VE* | *gmax* | *VP* |
|---|---|---|---|---|---|---|
| **Schedule** | | | | | | |
| $T$ | 1 | 1 | 1 | T | 1 | 1 |
| $f(t)$ | 0 | 0 | $-\frac{1}{2}\beta_0$ | 0 | 0 | $-\frac{1}{2}\beta_0 - \frac{1}{2}\beta_d t$ |
| $g^2(t)$ | $\sigma^2$ | $(\eta_1 - \eta_0|2t-1|)^2$ | $\beta_0$ | $2t$ | $\beta_0 + \beta_d t$ | $\beta_0 + \beta_d t$ |
| $\alpha_t$ | 1 | 1 | $e^{-\frac{1}{2}\beta_0 t}$ | 1 | 1 | $e^{-\frac{1}{2}\beta_0 t - \frac{1}{4}\beta_d t^2}$ |
| $\sigma_t^2$ | $\sigma^2 t$ | $\int_0^t g^2(\tau)\mathrm{d}\tau$ | $1 - e^{-\beta_0 t}$ | $t^2$ | $\beta_0 t + \frac{1}{2}\beta_d t^2$ | $1 - e^{-\beta_0 t - \frac{1}{2}\beta_d t^2}$ |
| $\bar{\alpha}_t$ | 1 | 1 | $e^{\frac{1}{2}\beta_0 - \frac{1}{2}\beta_0 t}$ | 1 | 1 | $\alpha_t/\alpha_1$ |
| $\rho_t^2$ | $\sigma^2 t$ | $\int_0^t g^2(\tau)\mathrm{d}\tau$ | $e^{\beta_0 t} - 1$ | $t^2$ | $\beta_0 t + \frac{1}{2}\beta_d t^2$ | $e^{\beta_0 t + \frac{1}{2}\beta_d t^2} - 1$ |
| $\bar{\rho}_t^2$ | $\sigma^2(1-t)$ | $\rho_1^2 - \rho_t^2$ | $e^{\beta_0} - e^{\beta_0 t}$ | $T^2 - t^2$ | $\rho_1^2 - \rho_t^2$ | $\rho_1^2 - \rho_t^2$ |
| **Parameters** | $\sigma$ | $\eta_0 = \frac{\sqrt{\beta_1}-\sqrt{\beta_0}}{2}$ $\eta_1 = \frac{\sqrt{\beta_1}+\sqrt{\beta_0}}{2}$ $\beta_0 = 0.1$ $\beta_1 = 0.3/1.0$ | $\beta_0$ | $T = 80$ | $\beta_0 = 0.01$ $\beta_d = 49.99$ | $\beta_0 = 0.01$ $\beta_d = 19.99$ |
| **Parameterization by Network $F_\theta$** | | | | | | |
| Data Predictor $x_\theta$ | Dependent on Training | $x_t - \sigma_t F_\theta$ | $c_{\text{skip}}(t)x_t + c_{\text{out}}(t)F_\theta$ | | $F_\theta$ | |

† Though I2SB is built on a discrete-time schedule for $T = 1000$ timesteps, it can be converted to a continuous-time schedule on $t \in [0, 1]$ approximately by mapping $t$ to $t/(T-1)$.

‡ The authors change to the same VP schedule as Bridge-TTS with parameters $\beta_0 = 0.1, \beta_d = 2$ in a revised version of their paper.

**Network Parameterization & Precondition** In practice, the neural network $F_\theta$ in DBMs does not always directly regress to the target score function; instead, it can predict other equivalent quantities, such as the *data predictor* $x_\theta = \frac{x_t - a_t x_T + c_t^2 s_\theta}{b_t}$ for a Gaussian $\mathcal{N}(a_t x_T + b_t x_0, c_t^2 I)$ like $q_{t|0T}$. Meanwhile, the inputs and outputs of the network $F_\theta$ could be rescaled for a better-behaved optimization process, known as the network precondition. As shown in Table 1, we could consistently use $x_0$ as the prediction target with different choices of network precondition to unify the previous practical designs for DBMs.

**PF-ODE and ODE Solver** The validity of a consistency model relies on an underlying PF-ODE that shares the same marginal distribution with the forward process. In the original DDBM paper [72], the marginal preserving property of the proposed ODE is justified following an analogous logic from the derivation of the PF-ODE of diffusion models [60] with Kolmogorov forward equation. However, its validity suffers from doubts as there is a singularity at the deterministic starting point $x_T$. Here, we provide a simple example to show that the ODE can indeed maintain the marginal distribution as long as we use a valid stochastic step to skip the singular point and start from $T - \gamma$ for any $\gamma > 0$.

**Example 3.1.** *Assume $T = 1$ and consider a simple Brownian Bridge between two fixed points $(x_0, x_1)$:*

$$\mathrm{d}x_t = \frac{x_1 - x_t}{1 - t}\mathrm{d}t + \mathrm{d}w_t, \tag{13}$$

*with marginal distribution $q_{t|01}(x_t|x_0, x_1) = \mathcal{N}((1-t)x_0 + tx_1, t(1-t))$. The ground-truth reverse SDE and PF-ODE are given by:*

$$\mathrm{d}x_t = \frac{x_t - x_0}{t}\mathrm{d}t + \mathrm{d}\bar{w}_t, \tag{14}$$

$$\mathrm{d}x_t = \left(\frac{1 - 2t}{2t(1-t)}x_t + \frac{1}{2(1-t)}x_1 - \frac{1}{2t}x_0\right)\mathrm{d}t. \tag{15}$$

*Then first simulating the reverse SDE in Eqn. (14) from $t = 1$ to $t = 1 - \gamma$ for some $\gamma \in (0, 1)$ and then starting to simulate the PF-ODE in Eqn. (15) will preserve the marginal distribution.*

The detailed derivation can be found in Appendix. B.2. Therefore, the time horizon of the consistency model based on the bridge ODE needs to be set as $t \in [\epsilon, T - \gamma]$ for some pre-specified $\epsilon, \gamma > 0$. Additionally, the marginal preservation of the bridge ODE for more general diffusion bridges can be strictly justified by considering non-Markovian variants, as done in DBIM [69].

Another crucial element for developing consistency models is the ODE solver, as a solver with a lower local error would yield lower error for consistency distillation, as well as the corresponding

consistency training objectives [58, 57]. Inspired by the successful practice of advanced ODE solvers based on the Exponential Integrator (EI) [4, 22] in diffusion models, we present a first-order bridge ODE solver in a similar fashion:

**Proposition 3.1.** *Given an initial value $\boldsymbol{x}_t$ at time $t > 0$, the first-order solver of the bridge ODE in Eqn. (8) from $t$ to $r \in [0, t]$ with the noise schedule defined in Eqn. (11) is:*

$$\boldsymbol{x}_r = \frac{\alpha_r \rho_r \bar{\rho}_r}{\alpha_t \rho_t \bar{\rho}_t} \boldsymbol{x}_t + \frac{\alpha_r}{\rho_T^2} \left[ \left( \bar{\rho}_r^2 - \frac{\bar{\rho}_t \rho_r \bar{\rho}_r}{\rho_t} \right) \boldsymbol{x}_{\boldsymbol{\theta}}(\boldsymbol{x}_t, t, \boldsymbol{y}) + \left( \rho_r^2 - \frac{\rho_t \rho_r \bar{\rho}_r}{\bar{\rho}_t} \right) \frac{\boldsymbol{y}}{\alpha_T} \right]. \qquad (16)$$

We provide detailed derivation in the Appendix B.1. Typically, an EI-based solver enjoys a lower discretization error and therefore has better empirical performance [16, 38, 39, 67, 70]. Another notable advantage of this general form solver, as we will show in Section 3.3, is that it could naturally establish the connection between consistency training and consistency distillation for any noise schedules that take the form in Eqn. (11), eliminating the dependence of the VE schedule and the corresponding Euler ODE solver in the common derivation [58].

## 3.2 Consistency Bridge Distillation

Analogous to consistency distillation with the empirical diffusion ODE, we could leverage a pretrained score predictor $\boldsymbol{s}_{\boldsymbol{\phi}}(\boldsymbol{x}_t, t, \boldsymbol{y}) \approx \nabla_{\boldsymbol{x}_t} \log q_{t|T}(\boldsymbol{x}_t | \boldsymbol{x}_T = \boldsymbol{y})$ to solve the empirical bridge ODE to obtain $\hat{\boldsymbol{x}}_r$, i.e., $\hat{\boldsymbol{x}}_r = \hat{\boldsymbol{x}}_{\boldsymbol{\phi}}(\boldsymbol{x}_t, t, r, \boldsymbol{y})$, where $\hat{\boldsymbol{x}}_{\boldsymbol{\phi}}$ is the update function of a one-step ODE solver with fixed $\boldsymbol{s}_{\boldsymbol{\phi}}$. We define the *consistency bridge distillation* (CBD) loss as:

$$\mathcal{L}_{\text{CBD}}^{\Delta t_{\max}} := \qquad (17)$$
$$\mathbb{E}_{t \in \mathcal{U}(\epsilon, T - \gamma), r = r(t)} \mathbb{E}_{q_{\text{data}}(\boldsymbol{x}, \boldsymbol{y}) q_{t|0T}(\boldsymbol{x}_t | \boldsymbol{x}_0 = \boldsymbol{x}, \boldsymbol{x}_T = \boldsymbol{y})} \left[ \lambda(t) d \left( \boldsymbol{h}_{\boldsymbol{\theta}}(\boldsymbol{x}_t, t, \boldsymbol{y}), \boldsymbol{h}_{\boldsymbol{\theta}^-}(\hat{\boldsymbol{x}}_{\boldsymbol{\phi}}(\boldsymbol{x}_t, t, r, \boldsymbol{y}), r, \boldsymbol{y}) \right) \right],$$

where $t$ is sampled from the uniform distribution over $[\epsilon, T - \gamma]$, $r(t)$ is a function specifies another timestep $r$ such that $\epsilon \leq r < t$ with $\Delta t_{\max} := \max_t \{t - r(t)\}$ and $\Delta t_{\min} := \min_t \{t - r(t)\}$, $\lambda(t)$ is a positive weighting function, $d$ is some distance metric function with $\forall \boldsymbol{x}, \boldsymbol{y} : d(\boldsymbol{x}, \boldsymbol{y}) \geq 0$ and $d(\boldsymbol{x}, \boldsymbol{y}) = 0$ iff. $\boldsymbol{x} = \boldsymbol{y}$, and $\boldsymbol{\theta}^- = \text{stopgrad}(\boldsymbol{\theta})$. Similarly to the case of consistency distillation in empirical diffusion ODEs, we have the following asymptotic analysis of the CBD objective:

**Proposition 3.2.** *Given $\Delta t_{\max} = \max_t \{t - r(t)\}$ and let $\boldsymbol{h}_{\boldsymbol{\phi}}(\cdot, \cdot, \cdot)$ be the consistency function of the empirical bridge ODE taking the form in Eqn. (8). Assume $\boldsymbol{h}_{\boldsymbol{\theta}}$ is a Lipschitz function, i.e., there exists $L > 0$, such that for all $t \in [\epsilon, T - \gamma], \boldsymbol{x}_1, \boldsymbol{x}_2, \boldsymbol{y}$, we have $\|\boldsymbol{h}_{\boldsymbol{\theta}}(\boldsymbol{x}_1, t, \boldsymbol{y}) - \boldsymbol{h}_{\boldsymbol{\theta}}(\boldsymbol{x}_2, t, \boldsymbol{y})\|_2 \leq L \|\boldsymbol{x}_1 - \boldsymbol{x}_2\|_2$. Meanwhile, assume that for all $t, r \in [\epsilon, T - \gamma], \boldsymbol{y} \sim q_{\text{data}}(\boldsymbol{y}) := \mathbb{E}_{\boldsymbol{x}}[q_{\text{data}}(\boldsymbol{x}, \boldsymbol{y})]$, the ODE solver $\hat{\boldsymbol{x}}_{\boldsymbol{\phi}}(\cdot, t, r, \boldsymbol{y})$ has local error uniformly bounded by $O((t - r)^{p+1})$ with $p \geq 1$. Then, if $\mathcal{L}_{\text{CBD}}^{\Delta t_{\max}} = 0$, we have: $\sup_{t, \boldsymbol{x}, \boldsymbol{y}} \|\boldsymbol{h}_{\boldsymbol{\theta}}(\boldsymbol{x}, t, \boldsymbol{y}) - \boldsymbol{h}_{\boldsymbol{\phi}}(\boldsymbol{x}, t, \boldsymbol{y})\|_2 = O((\Delta t_{\max})^p)$.*

The vast majority of the analysis can be done by directly following the proof in [58] with minor differences between the overlapped timestep intervals $\{t, r(t)\}$ for $t \in [\epsilon, T - \gamma]$ used in Eqn. (17) and the fixed timestep intervals $\{t_n\}_{n=1}^N$ used in [58]. We include it in Appendix B.5 for completeness. In this work, unless otherwise stated, we use the first-order ODE solver in Eqn. (16) as $\hat{\boldsymbol{x}}_{\boldsymbol{\phi}}$.

## 3.3 Consistency Bridge Training

In addition to distilling from pre-trained score predictor $\boldsymbol{s}_{\boldsymbol{\phi}}$, consistency models can be trained [58, 57] or fine-tuned [15] by maintaining only one set of parameters $\boldsymbol{\theta}$. To accomplish this, we could leverage the unbiased score estimator:

$$\nabla_{\boldsymbol{x}_t} \log q_{t|T}(\boldsymbol{x}_t | \boldsymbol{x}_T = \boldsymbol{y}) = \mathbb{E}_{\boldsymbol{x}_0}[\nabla_{\boldsymbol{x}_t} \log q_{t|0T}(\boldsymbol{x}_t | \boldsymbol{x}_0, \boldsymbol{x}_T) | \boldsymbol{x}_t, \boldsymbol{x}_T = \boldsymbol{y}], \qquad (18)$$

that is, with a single sample $(\boldsymbol{x}, \boldsymbol{y}) \sim q_{\text{data}}$ and $\boldsymbol{x}_t \sim q_{t|0T}(\boldsymbol{x}_t | \boldsymbol{x}_0 = \boldsymbol{x}, \boldsymbol{x}_T = \boldsymbol{y})$, the score $\nabla_{\boldsymbol{x}_t} \log q_{t|T}(\boldsymbol{x}_t | \boldsymbol{x}_T = \boldsymbol{y})$ can be estimated with $\nabla_{\boldsymbol{x}_t} \log q_{t|0T}(\boldsymbol{x}_t | \boldsymbol{x}_0, \boldsymbol{x}_T)$. Substituting such an estimation of $\boldsymbol{s}_{\boldsymbol{\phi}}$ into the one-step ODE solver $\hat{\boldsymbol{x}}_{\boldsymbol{\phi}}$ in Eqn. (17) with the transformation between data and score predictor $\boldsymbol{x}_{\boldsymbol{\phi}} = \frac{\boldsymbol{x}_t - a_t \boldsymbol{x}_T + c_t^2 \boldsymbol{s}_{\boldsymbol{\phi}}}{b_t}$, we can obtain an alternative $\hat{\boldsymbol{x}}_r$ that does not rely on the pre-trained $\boldsymbol{s}_{\boldsymbol{\phi}}$ for any noise schedule taking the form in Eqn. (11) as follows (detail in Appendix B.3):

$$\hat{\boldsymbol{x}}_r = \hat{\boldsymbol{x}}(\boldsymbol{x}_t, t, r, \boldsymbol{x}, \boldsymbol{y}) = a_r \boldsymbol{y} + b_r \boldsymbol{x} + c_r \boldsymbol{z}, \qquad (19)$$

where $a_r, b_r, c_r$ are defined as in Eqn. (11), and $z = \frac{x_t - a_t y - b_t x}{c_t} \sim \mathcal{N}(0, I)$. Based on this instantiation of $\hat{x}_r$, we define the *consistency bridge training* (CBT) loss as:

$$\mathcal{L}_{\text{CBT}}^{\Delta t_{\max}} := \tag{20}$$
$$\mathbb{E}_{t \in \mathcal{U}(\epsilon, T - \gamma), r = r(t)} \mathbb{E}_{q_{\text{data}}(x, y)} \left[ \lambda(t) d \left( h_\theta(a_t y + b_t x + c_t z, t, y), h_{\theta^-}(a_r y + b_r x + c_r z, r, y) \right) \right],$$

where $t, r(\cdot), \lambda(\cdot), \theta^{-1}$ are defined the same as in Eqn. (17), and $z \sim \mathcal{N}(0, I)$ is a shared Gaussian noise used in both $h_\theta$ and $h_{\theta^{-1}}$. We have the following proposition demonstrating the connection between $\mathcal{L}_{\text{CBT}}^{\Delta t_{\max}}$ and $\mathcal{L}_{\text{CBD}}^{\Delta t_{\max}}$ with the first-order one-step ODE solver:

**Proposition 3.3.** *Given $\Delta t_{\max} = \max_t \{t - r(t)\}$ and assume $d, h_\theta, f, g$ are twice continuously differentiable with bounded second derivatives, the weighting function $\lambda(\cdot)$ is bounded, and $\mathbb{E}[\|\nabla_{x_t} \log q_{t|T}(x_t|x_T)\|_2^2] < \infty$. Meanwhile, assume that $\mathcal{L}_{\text{CBD}}^{\Delta t_{\max}}$ employs the one-step ODE solver in Eqn. (16) with ground truth pre-trained score model, i.e., $\forall t \in [\epsilon, T - \gamma], y \sim q_{\text{data}}(y)$: $s_\phi(x_t, t, y) \equiv \nabla_{x_t} \log q_{t|T}(x_t|x_T = y)$. Then, we have: $\mathcal{L}_{\text{CBD}}^{\Delta t_{\max}} = \mathcal{L}_{\text{CBT}}^{\Delta t_{\max}} + o(\Delta t_{\max})$.*

The core part of our analysis also follows [58], except the connection between the CBD & CBT objective relies on the proposed first-order ODE solver and the estimated $\hat{x}_r$ in Eqn. (19) with the general noise schedule for DDBM. We include the details in Appendix B.6.

### 3.4 Network Precondition and Sampling

**Network Precondition**    First, we focus on enforcing the boundary condition $h_\theta(x_\epsilon, \epsilon, y) = x_\epsilon$ of our consistency bridge model, which can be done by designing a proper network precondition. Usually, a variable substitution $\tilde{t} = t - \epsilon$ could work in most cases. For example, for the precondition for I²SB in Table 1, we have $x_\epsilon + \sigma_{\tilde{\epsilon}} F_\theta = x_\epsilon + \sqrt{\int_0^{\epsilon - \epsilon} g^2(\tau) d\tau} = x_\epsilon$. Also, the common "EDM" [24] style precondition used in DDBM also satisfies $c_{\text{skip}}(\tilde{\epsilon}) = 1$ and $c_{\text{out}}(\tilde{\epsilon}) = 0$. We also give a universal precondition to satisfy the boundary conditions based on the form of the ODE solver in Eqn. (16) in Appendix B.4 to cope with the case where the variable substitution is not applicable.

**Sampling**    As explained in Section 3.1, the PF-ODE is only well-defined within the time horizon $0 \le t \le T - \gamma$ for some $\gamma \in (0, T)$. Hence, the sampling of CDBMs should start with $x_{T-\gamma} \sim q_{T-\gamma|T}(x_{T-\gamma}|x_T = y)$, which can be obtained by simulating the reverse SDE in Eqn. (7) from $T$ to $T - \gamma$. Here we opt to use one first-order stochastic step, which is equivalent to performing posterior sampling, i.e., $x_{T-\gamma} \sim q_{T-\gamma|0T}(x_{T-\gamma}|x_0 = h_\theta(x_T, T, y), x_T = y)$. This sampling approach defaults to two NFEs (Number of Function Evaluations), which is aligned with the practical guideline that employing two-step sampling in CM allows for a better trade-off between quality and computation compared to other treatments such as scaling up models [15]. We could also alternate a forward noising step and a backward consistency step multiple times to further improve sample quality as consistency models do.

## 4 Experiments

### 4.1 Experimental Setup

**Task, Datasets, and Metrics**    In this work, we conduct experiments for CDBM on image-to-image translation and image inpainting tasks with various image resolutions and scales of the data set. For image-to-image translation, we use the Edges→Handbags [23] with $64 \times 64$ pixel resolution and DIODE-Outdoor [62] with $256 \times 256$ pixel resolution. For image inpainting, we choose ImageNet [9] $256 \times 256$ with a center mask of size $128 \times 128$. Regarding the evaluation metrics, we report the Fréchet inception distance (FID) [19] for all datasets. Furthermore, following previous works [33, 72], we measure Inception Scores (IS) [3], LPIPS [68] and Mean Square Error (MSE) for image-to-image translation and Classifier Accuracy (CA) of a pre-trained ResNet50 for image-inpainting. The metrics are computed using the complete training set for Edges→Handbags and DIODE-Outdoor, and a validation subset of 10,000 images for ImageNet.

**Training Configurations**    We train CDBM in two ways: distill pre-trained DDBM with CBD or *fine-tuning* DDBM with CBT. We keep the noise schedule and prediction target of the pre-trained DDBM

unchanged and modify the network precondition to satisfy the boundary condition. Specifically, we adopt the design space of DDBM-VP and I$^2$SB in Table 1 on image-to-image translation and image inpainting, respectively. We specify complete training details in Appendix C.

**Specification of Design Choices**  We illustrate the specific design choices for CDBM. In this work, we use $t \in [\epsilon, 1-\gamma]$ and set $\epsilon = 0.0001, \gamma = 0.001$ and sample $t$ uniformly during training. We employ two different sets of the timestep function $r(t)$ and the loss weighting $\lambda(t)$, also named the *training schedule* for CDBM. The first, following [58], specifies a constant quantity for $\Delta t = t - r(t)$ with a simple loss weighting of $\lambda(t) = 1$. The constant gap $\Delta t$ is treated as a hyperparameter and we search it among $\{1/9, 1/18, 1/36, 1/60, 1/80, 1/120\}$. The other employs $r(t)$ that gradually shrinks $t - r(t)$ during the training process and a loss weighting of $\lambda(t) = \frac{1}{t-r(t)}$, which enjoys a better trade-off between faster convergence and performance [58, 57, 15]. Following [15], we use a sigmoid-style function $r(t) = t(1 - \frac{1}{q^{\lfloor \text{iters}/s \rfloor}})(1 + \frac{k}{1+e^{bt}})$, where iters is the number of training iterations, $q, s, k, b$ are hyperparameters. We use $q = 2, k = 8$, and tune $b \in \{1, 2, 5, 10, 20, 50\}$ and $s \in \{5000, 10000\}$.

Table 2: Quantitative Results on the Image-to-Image Translation Task

| | Edges→Handbags ($64 \times 64$) | | | | DIODE-Outdoor ($256 \times 256$) | | | |
|---|---|---|---|---|---|---|---|---|
| | FID ↓ | IS ↑ | LPIPS ↓ | MSE ↓ | FID ↓ | IS ↑ | LPIPS ↓ | MSE ↓ |
| Pix2Pix [23] | 74.8 | 4.24 | 0.356 | 0.209 | 82.4 | 4.22 | 0.556 | 0.133 |
| DDIB [61] | 186.84 | 2.04 | 0.869 | 1.05 | 242.3 | 4.22 | 0.798 | 0.794 |
| SDEdit [41] | 26.5 | 3.58 | 0.271 | 0.510 | 31.14 | 5.70 | 0.714 | 0.534 |
| Rectified Flow [35] | 25.3 | 2.80 | 0.241 | 0.088 | 77.18 | 5.87 | 0.534 | 0.157 |
| I$^2$SB [33] | 7.43 | 3.40 | 0.244 | 0.191 | 9.34 | 5.77 | 0.373 | 0.145 |
| DDBM [72] (NFE=118) | 1.83 | 3.73 | 0.142 | 0.0402 | 4.43 | 6.21 | 0.244 | 0.0839 |
| DDBM (ODE-1, NFE=2) | 6.70 | 3.71 | 0.0968 | 0.0037 | 73.08 | 6.67 | 0.318 | 0.0118 |
| DDBM (ODE-1, NFE=50) | 1.14 | 3.62 | 0.0979 | 0.0054 | 3.20 | 6.08 | 0.198 | 0.0179 |
| DDBM (ODE-1, NFE=100) | 0.89 | 3.62 | 0.0995 | 0.0056 | 2.57 | 6.06 | 0.198 | 0.0183 |
| CBD (Ours, NFE=2) | 1.30 | 3.62 | 0.128 | 0.0124 | 3.66 | 6.02 | 0.224 | 0.0216 |
| CBT (Ours, NFE=2) | **0.80** | 3.65 | 0.106 | 0.0068 | 2.93 | 6.06 | 0.205 | 0.0181 |

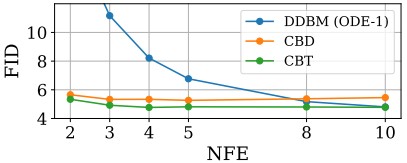

Figure 2: NFE-FID plot of CDBM and DDBM on ImageNet $256 \times 256$

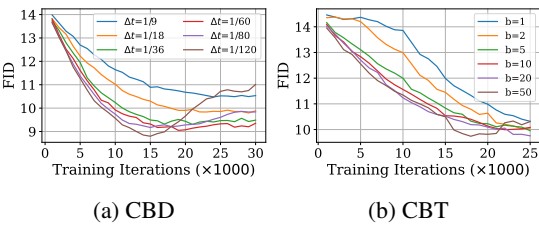

(a) CBD          (b) CBT

Figure 3: Ablation for hyperparameters of CDBM

Table 3: Quantitative Results on the Image Inpainting Task

| ImageNet ($256 \times 256$) Center mask $128 \times 128$ | FID ↓ | CA ↑ |
|---|---|---|
| DDRM [25] | 24.4 | 62.1 |
| ΠGDM [56] | 7.3 | 72.6 |
| DDNM [65] | 15.1 | 55.9 |
| Palette [49] | 6.1 | 63.0 |
| CDSB [52] | 50.5 | 49.6 |
| I$^2$SB [33] | 4.9 | 66.1 |
| DDBM (ODE-1, NFE=2) | 17.17 | 59.6 |
| DDBM (ODE-1, NFE=10) | 4.81 | 70.7 |
| CBD (Ours, NFE=2) | 5.65 | 69.6 |
| CBD (Ours, NFE=4) | 5.34 | 69.6 |
| CBT (Ours, NFE=2) | 5.34 | 69.8 |
| CBT (Ours, NFE=4) | **4.77** | 70.3 |

## 4.2  Results for Few-step Generation

We present the quantitative results of CDBM on image-to-image translation and image inpainting tasks in Table 2 and Table 3. We adopt DDBM on the same noise schedule and network architecture, with the first-order ODE solver in Eqn. (16) as our main baseline (i.e., "DDBM (ODE-1)"). We report the performance of the baseline DDBM under different Number of Function Evaluations (NFE) as a

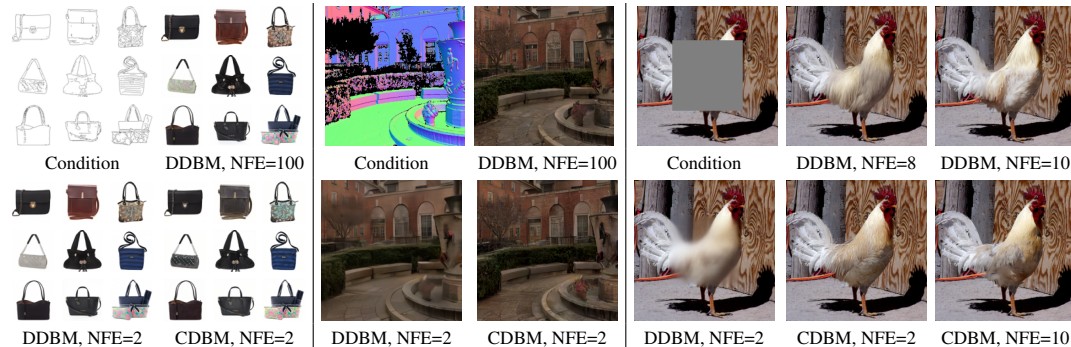

Figure 4: Qualitative demonstration between DDBM and CDBM.

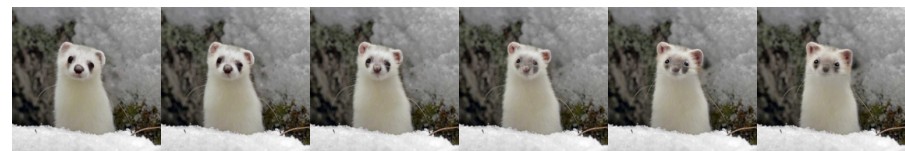

Figure 5: Example semantic interpolation result with CDBMs

reference for the sampling acceleration ratio (Reduction factor of NFE to achieve the same FID) of CDBM. Following [72, 33], we report the result of other baselines with NFE $\geq 40$, which consists of diffusion-based methods, diffusion bridges with different formulations, or samplers. We mainly focus on the two-step generation scenario for CDBM, which is the minimal NFEs required for CDBM using the sampling procedure described in Section 3.4.

For image-to-image translation, as shown in Table. 2, we first observed that our proposed first-order ODE solver has superior performance compared to the hybrid high-order sampler used in DDBM [72]. On top of that, CDBM's FID at NFE $= 2$ is close to or even better than DDBM's at NFE around $100$ with the advanced ODE solver, achieving a sampling speed-up around $50\times$. This can be corroborated by the qualitative demonstration in Fig. 4, where CDBMs drastically reduce the blurring effect on DDBMs under few-step generation settings while enjoying realistic and faithful translation performance.

For image inpainting, as shown in Table. 3, the baseline ODE solver for DDBM achieves decent sample quality at NFE $= 10$. For CDBM, as shown in Fig. 2, the acceleration ratio is relatively modest in such a large-scale and challenging dataset, achieving close to a $4\times$ increase in sampling speed. Notably, CBT's FID at NFE $= 4$ matches DDBM at NFE $= 10$. Moreover, we find that CDBMs have better visual quality than DDBM given the same computation budget, as shown in Fig. 4 and Appendix D, which illustrates that CDBM yields a better quality-efficiency trade-off.

Meanwhile, we observe that fine-tuning DDBMs with CBT generally produces better results than CBD in all three data sets, demonstrating fine-tuning a pre-trained score model to a consistency function is a more promising solution with less computational and memory cost compared to distillation, which is consistent with recent findings [15]. We also conducted an ablation study for CBD and CBT under different training schedules (i.e., the combination of the timestep function $r(t)$ and the loss weighting $\lambda(t)$) on ImageNet $256 \times 256$. As shown in Fig. 3, for a small timestep interval $t - r(t)$, e.g., a small $\Delta t$ in Fig. 3a or a large $b$ in Fig. 3b (detail in Appendix C.2), the performance is generally better but also suffers from training instability, indicated by the sharp increase in FID during training when $\Delta t = 1/120$ and $b = 50$. While for a large timestep interval, the performance at convergence is usually worse. In practice, we found that adopting the training schedule that gradually shrinks $r(t) - t$ with $b = 20$ or $50$ with CBT could work across all tasks, whereas CBD generally needs a meticulous design for $\Delta t$ or $b$ to ensure stable training and satisfactory performance.

### 4.3 Semantic Interpolation

We show that CDBMs support performing downstream tasks, such as semantic interpolation, similar to diffusion models [55]. Recall that the sampling process for CDBM alternates between consistency

function evaluation and forward sampling, we could track all noises and the corresponding timesteps to re-generate the same sample. By interpolating the noises of two sampling trajectories, we can obtain a series of samples lying between the semantics of two source samples, as shown in Fig. 5, which demonstrates that CDBMs have a wide range of generative modeling capabilities, such as sample diversity and semantic interpolation.

## 5 Conclusion

In this work, we introduce consistency diffusion bridge models (CDBMs) to address the sampling inefficiency of DDBMs and present two frameworks, consistency bridge distillation and consistency bridge training, to learn the consistency function of the DDBM's PF-ODE. Building on a unified view of design spaces and the corresponding general-form ODE solver, CDBM exhibits significant flexibility and adaptability, allowing for straightforward integration with previously established successful designs for diffusion bridges. Experimental evaluations across three datasets show that CDBM can effectively boost the sampling speed of DDBM by $4\times$ to $50\times$. Furthermore, it achieves the saturated performance of DDBMs with less than five NFEs and possesses the broad capacity of generative models, such as sample diversity and semantic interpolation.

**Limitations and Broader Impact** While significantly improving the sampling efficiency in the datasets we used, it remains to be explored how the proposed CDBM, along with the DDBM formulation, performs in datasets with larger-scale or more complex characteristics. Furthermore, the consistency model paradigm typically suffers from numerical instability and it would be a promising research direction to keep improving CDBM's performance from an optimization perspective. With enhanced sampling efficiency, CDBMs could contribute to more energy-efficient deployment of generative models, aligning with broader goals of sustainable AI development. However, it could also lower the cost associated with the potential misuse for creating deceptive content. We hope that our work will be enforced with certain ethical guidelines to prevent any form of harm.

## Acknowledgments and Disclosure of Funding

This work was supported by the National Science and Technology Major Project (2021ZD0110502), NSFC Projects (Nos. 62350080, 62106122, 92248303, 92370124, 62350080, 62276149, U2341228, 62076147), Tsinghua Institute for Guo Qiang, and the High Performance Computing Center, Tsinghua University. J. Zhu was also supported by the XPlorer Prize.

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

# A  Related Works

**Diffusion Bridges**  Diffusion bridges [44, 36, 33, 54, 51, 72, 7, 6] are an emerging class of generative models with attractive flexibility in modeling the stochastic process between two arbitrary distributions. The flow matching [32], and its stochastic counterpart, bridge matching [44] assume the access of a joint distribution and an interpolation, or a forward process, between the samples, then, another SDE/ODE is learned to estimate the dynamics of the pre-defined interpolation, which can be used for generative modeling from non-Gaussian priors [33, 6, 72, 69, 66]. In particular, the forward process can be constructed via Doob's $h$-transform [44, 36, 72]. Among them, DDBM [72] focuses on learning the reverse-time diffusion bridge conditioned on a particular terminal endpoint with denoising score matching, which has been shown to be equivalent to conducting a conditioned bridge matching that preserves the initial joint distribution [7]. Other works tackle solving the diffuion Schrödinger Bridge problem, such as using iterative algorithms [8, 51, 43]. In this work, we use a unified view of design spaces on existing diffusion bridges, in particular, bridge matching methods, to decouple empirical choices from their different theoretical premises and properties and focus on developing the techniques of learning the consistency function of DDBM's PF-ODE with various established design choices for diffusion bridges.

**Consistency Models**  Recent studies have continued to explore the effectiveness of consistency models [58]. For example, CTM [26] proposes to augment the prediction target from the starting point to the intermediate points along the PF-ODE trajectory from the input to this starting point. BCM [31] additionally expands the model to allow direct mapping at the PF-ODE trajectory points in both forward and reverse time. Beyond different formulations, several works aim to improve the performance of consistency training with theoretical and practical insights. iCT [57] systematically examines the design choices of consistency training and presents improved training schedule, loss weighting, distance metrics, etc. ECT [15] further leverages the insights to propose novel practical designs and show fine-tuning pre-trained diffusion models for learning consistency models yields decent performance with much lower computation compared to distillation. Unlike these works, we focus on constructing consistency models on top of the formulation of DDBMs with specialized design spaces and a sophisticated ODE solver for them.

# B  Additional Details for CDBM Formulation, CBD, and CBT

## B.1  Derivation of First-Order Bridge ODE Solver

We first review the first-order ODE solver in Section 3.1:

**Proposition 3.1.** *Given an initial value $\boldsymbol{x}_t$ at time $t > 0$, the first-order solver of the bridge ODE in Eqn. (8) from $t$ to $r \in [0, t]$ with the noise schedule defined in Eqn. (11) is:*

$$\boldsymbol{x}_r = \frac{\alpha_r \rho_r \bar{\rho}_r}{\alpha_t \rho_t \bar{\rho}_t} \boldsymbol{x}_t + \frac{\alpha_r}{\rho_T^2} \left[ \left( \bar{\rho}_r^2 - \frac{\bar{\rho}_t \rho_r \bar{\rho}_r}{\rho_t} \right) \boldsymbol{x}_{\boldsymbol{\theta}}(\boldsymbol{x}_t, t, \boldsymbol{y}) + \left( \rho_r^2 - \frac{\rho_t \rho_r \bar{\rho}_r}{\bar{\rho}_t} \right) \frac{\boldsymbol{y}}{\alpha_T} \right]. \qquad (16)$$

Recall the PF-ODE of DDBM in Eqn. (8) with a linear drift $f(t)\boldsymbol{x}_t$:

$$\mathrm{d}\boldsymbol{x}_t = \left[ f(t)\boldsymbol{x}_t - g^2(t) \left[ \frac{1}{2} \nabla_{\boldsymbol{x}_t} \log q_{t|T}(\boldsymbol{x}_t | \boldsymbol{x}_T = \boldsymbol{y}) - \nabla_{\boldsymbol{x}_t} \log p_{T|t}(\boldsymbol{x}_T = \boldsymbol{y} | \boldsymbol{x}_t) \right] \right] \mathrm{d}t. \qquad (21)$$

Also recall the noise schedule in Eqn. (11) and the analytic form of $p_{t|0}$ and $p_{T|t}$ in diffusion models:

$$p_{t|0}(\boldsymbol{x}_t | \boldsymbol{x}_0) = \mathcal{N}\left( \alpha_t \boldsymbol{x}_0, \alpha_t^2 \rho_t^2 \boldsymbol{I} \right), \quad p_{T|t}(\boldsymbol{x}_T | \boldsymbol{x}_t) = \mathcal{N}\left( \frac{\alpha_T}{\alpha_t} \boldsymbol{x}_t, \alpha_T^2 (\rho_T^2 - \rho_t^2) \boldsymbol{I} \right),$$

$$q_{t|0T}(\boldsymbol{x}_t | \boldsymbol{x}_0, \boldsymbol{x}_T) = p_{t|0T}(\boldsymbol{x}_t | \boldsymbol{x}_0, \boldsymbol{x}_T) = \mathcal{N}\left( a_t \boldsymbol{x}_T + b_t \boldsymbol{x}_0, c_t^2 \boldsymbol{I} \right), \qquad (22)$$

$$\text{where} \quad a_t = \frac{\bar{\alpha}_t \rho_t^2}{\rho_T^2}, \quad b_t = \frac{\alpha_t \bar{\rho}_t^2}{\rho_T^2}, \quad c_t^2 = \frac{\alpha_t^2 \bar{\rho}_t^2 \rho_t^2}{\rho_T^2}.$$

We thus have the corresponding score functions and the score-data transformation for $\boldsymbol{s}_{\boldsymbol{\theta}}$ that predicts $\nabla_{\boldsymbol{x}_t} \log q_{t|0T}$:

$$\nabla_{\boldsymbol{x}_t} \log p_{T|t}(\boldsymbol{x}_T = \boldsymbol{y} | \boldsymbol{x}_t) = -\frac{\boldsymbol{x}_t - \bar{\alpha}_t \boldsymbol{y}}{\alpha_t^2 \bar{\rho}_t^2}, \qquad (23)$$

$$\nabla_{\boldsymbol{x}_t} \log q_{t|0T}(\boldsymbol{x}_t|\boldsymbol{x}_0, \boldsymbol{x}_T = \boldsymbol{y}) = -\frac{\boldsymbol{x}_t - (\alpha_t \bar{\rho}_t^2 \boldsymbol{x}_0 + \bar{\alpha}_t \rho_t^2 \boldsymbol{x}_T)/\rho_T^2}{\alpha_t^2 \bar{\rho}_t^2 \rho_t^2 / \rho_T^2}, \tag{24}$$

$$\boldsymbol{s}_{\boldsymbol{\theta}}(\boldsymbol{x}_t, t, \boldsymbol{y}) = -\frac{\boldsymbol{x}_t - (\alpha_t \bar{\rho}_t^2 \boldsymbol{x}_{\boldsymbol{\theta}}(\boldsymbol{x}_t, t, \boldsymbol{y}) + \bar{\alpha}_t \rho_t^2 \boldsymbol{x}_T)/\rho_T^2}{\alpha_t^2 \bar{\rho}_t^2 \rho_t^2 / \rho_T^2}. \tag{25}$$

We use the data parameterization $\boldsymbol{x}_{\boldsymbol{\theta}}(\boldsymbol{x}_t, t, \boldsymbol{y})$ in following discussions. For PF-ODE in Eqn. (21), substituting $\nabla_{\boldsymbol{x}_t} \log q_{t|T}(\boldsymbol{x}_t|\boldsymbol{x}_T = \boldsymbol{y})$ with Eqn. (25) and substituting $p_{T|t}(\boldsymbol{x}_T|\boldsymbol{x}_t)$ in with Eqn. (23), we have the following after some simplification:

$$d\boldsymbol{x}_t = \left[ f(t)\boldsymbol{x}_t - \frac{1}{2}g^2(t)\frac{\boldsymbol{x}_t - \bar{\alpha}_t \boldsymbol{y}}{\alpha_t^2 \bar{\rho}_t^2} + \frac{1}{2}g^2(t)\frac{\boldsymbol{x}_t - \alpha_t \boldsymbol{x}_{\boldsymbol{\theta}}(\boldsymbol{x}_t, t, \boldsymbol{y})}{\alpha_t^2 \rho_t^2} \right] dt. \tag{26}$$

which shares the same form as the ODE in Bridge-TTS [6]. In the next discussions, we present an overview of deriving the first-order ODE solver and refer the reader to Appendix A.2 in [6] for details.

We begin by reviewing exponential integrators [4, 22], a key technique for developing advanced diffusion ODE solvers [16, 38, 39, 70]. Consider the following ODE:

$$d\boldsymbol{x}_t = [a(t)\boldsymbol{x}_t + b(t)\boldsymbol{F}_{\boldsymbol{\theta}}(\boldsymbol{x}_t, t)]dt, \tag{27}$$

where $\boldsymbol{F}_{\boldsymbol{\theta}}$ is a $n$-th differentiable parameterized function. By leveraging the "variation-of-constant" formula, we could obtain a specific form of the solution of the ODE in Eqn. (27) (assume $r < t$):

$$\boldsymbol{x}_r = e^{\int_t^r a(\tau)d\tau} \boldsymbol{x}_t + \int_t^r e^{\int_\tau^r a(s)ds} b(\tau) \boldsymbol{F}_{\boldsymbol{\theta}}(\boldsymbol{x}_\tau, \tau)d\tau, \tag{28}$$

The integral in Eqn. (28) only involves the function $\boldsymbol{F}_{\boldsymbol{\theta}}$, which helps reduce discretization errors.

With such a key methodology, we could derive the first-order solver for Eqn. (26). First, collecting the coefficients for $\boldsymbol{x}_t, \boldsymbol{y}, \boldsymbol{x}_{\boldsymbol{\theta}}$, we have:

$$d\boldsymbol{x}_t = \left[ \left( f(t) - \frac{g^2(t)}{2\alpha_t^2 \bar{\rho}_t^2} + \frac{g^2(t)}{2\alpha_t^2 \rho_t^2} \right) \boldsymbol{x}_t + \frac{g^2(t)\bar{\alpha}_t}{2\alpha_t^2 \bar{\rho}_t^2} \boldsymbol{y} - \frac{g^2(t)}{2\alpha_t \rho_t^2} \boldsymbol{x}_{\boldsymbol{\theta}}(\boldsymbol{x}_t, t, \boldsymbol{y}) \right] dt. \tag{29}$$

By setting:

$$a(t) = \left( f(t) - \frac{g^2(t)}{2\alpha_t^2 \bar{\rho}_t^2} + \frac{g^2(t)}{2\alpha_t^2 \rho_t^2} \right), \quad b_1(t) = \frac{g^2(t)\bar{\alpha}_t}{2\alpha_t^2 \bar{\rho}_t^2}, \quad b_2(t) = \frac{g^2(t)}{2\alpha_t \rho_t^2}.$$

with correspondence to Eqn. (28), the exponential terms could be analytically given by:

$$e^{\int_t^r a(\tau)d\tau} = \frac{\alpha_r \sigma_r \bar{\sigma}_r}{\alpha_t \sigma_t \bar{\sigma}_t}, \quad e^{\int_\tau^r a(s)ds} = \frac{\alpha_r \sigma_r \bar{\sigma}_r}{\alpha_\tau \sigma_\tau \bar{\sigma}_\tau}. \tag{30}$$

The exact solution for Eqn. (29) is thus given by:

$$\boldsymbol{x}_r = \frac{\alpha_r \rho_r \bar{\rho}_r}{\alpha_t \rho_t \bar{\rho}_t} \boldsymbol{x}_t + \frac{\bar{\alpha}_r \rho_r \bar{\rho}_r}{2} \int_t^r \frac{g^2(\tau)}{\alpha_\tau^2 \rho_\tau \bar{\rho}_\tau^3} \boldsymbol{y}d\tau - \frac{\alpha_r \rho_r \bar{\rho}_r}{2} \int_t^r \frac{g^2(\tau)}{\alpha_\tau^2 \rho_\tau^3 \bar{\rho}_\tau} \boldsymbol{x}_{\boldsymbol{\theta}}(\boldsymbol{x}_\tau, \tau)d\tau \tag{31}$$

The integrals in Eqn. (31) (without considering $\boldsymbol{x}_{\boldsymbol{\theta}}$) can be calculated as:

$$\int_t^r \frac{g^2(\tau)}{\alpha_\tau^2 \rho_\tau \bar{\rho}_\tau^3} d\tau = \frac{2}{\rho_T^2} \left( \frac{\rho_r}{\bar{\rho}_r} - \frac{\rho_t}{\bar{\rho}_t} \right), \quad \int_t^r \frac{g^2(\tau)}{\alpha_\tau^2 \sigma_\tau^3 \bar{\sigma}_\tau} d\tau = \frac{2}{\rho_T^2} \left( \frac{\bar{\rho}_t}{\rho_t} - \frac{\bar{\rho}_r}{\rho_r} \right)$$

Then, with the first order approximation $\boldsymbol{x}_{\boldsymbol{\theta}}(\boldsymbol{x}_\tau, \tau) \approx \boldsymbol{x}_{\boldsymbol{\theta}}(\boldsymbol{x}_s, s)$, we could obtain the first order solver in Eqn. (16).

## B.2 An Illustration Example of the Validity of the Bridge ODE

Recall the provided example in Section 3.1:

**Example 3.1.** *Assume $T = 1$ and consider a simple Brownian Bridge between two fixed points $(x_0, x_1)$:*

$$dx_t = \frac{x_1 - x_t}{1 - t} dt + dw_t, \tag{13}$$

*with marginal distribution $q_{t|01}(x_t|x_0, x_1) = \mathcal{N}((1-t)x_0 + tx_1, t(1-t))$. The ground-truth reverse SDE and PF-ODE are given by:*

$$\mathrm{d}x_t = \frac{x_t - x_0}{t}\mathrm{d}t + \mathrm{d}\bar{w}_t, \tag{14}$$

$$\mathrm{d}x_t = \left(\frac{1-2t}{2t(1-t)}x_t + \frac{1}{2(1-t)}x_1 - \frac{1}{2t}x_0\right)\mathrm{d}t. \tag{15}$$

*Then first simulating the reverse SDE in Eqn. (14) from $t = 1$ to $t = 1 - \gamma$ for some $\gamma \in (0, 1)$ and then starting to simulate the PF-ODE in Eqn. (15) will preserve the marginal distribution.*

*Proof.* We first demonstrate the effect of the initial SDE step, according to Table 1 and the expression of the relevant score terms in Eqn. (23) and Eqn. (25), the ground-truth reverse SDE can be derived as:

$$\mathrm{d}x_t = \frac{x_t - x_0}{t}\mathrm{d}t + \mathrm{d}\bar{w}_t.$$

Then, the analytic solution of the reverse SDE in Eqn. (7) from time $t$ to time $s < t$ can be derived as:

$$\mathrm{d}x_t - \frac{1}{t}x_t\mathrm{d}t = -\frac{1}{t}x_0 + \mathrm{d}\bar{w}_t$$

$$\Longleftrightarrow \mathrm{d}\left(\frac{1}{t}x_t\right) = -\frac{1}{t^2}x_0 + \frac{1}{t}\mathrm{d}\bar{w}_t$$

$$\Longleftrightarrow \frac{1}{s}x_s - \frac{1}{t}x_t = \left(\frac{1}{s} - \frac{1}{t}\right)x_0 + \sqrt{\frac{1}{s} - \frac{1}{t}}\epsilon, \quad \epsilon \sim \mathcal{N}(0, 1).$$

Let $t = 1$, we have:

$$x_s = (1 - s)x_0 + sx_1 + \sqrt{s(1 - s)}\epsilon,$$

i.e., $x_s$ has the same marginal as the forward process at time $s$. Similarly, the ground-truth PF-ODE can be derived as:

$$\mathrm{d}x_t = \left(\frac{1-2t}{2t(1-t)}x_t + \frac{1}{2(1-t)}x_1 - \frac{1}{2t}x_0\right)\mathrm{d}t,$$

whose analytic solution from time $t$ to time $s < t$ can be derived as:

$$\mathrm{d}x_t - \frac{1-2t}{2t(1-t)}x_t\mathrm{d}t = \frac{1}{2(1-t)}x_1\mathrm{d}t - \frac{1}{2t}x_0\mathrm{d}t$$

$$\Longleftrightarrow \mathrm{d}\left(\frac{1}{\sqrt{t(1-t)}}x_t\right) = \frac{t}{2[t(1-t)]^{3/2}}x_1\mathrm{d}t - \frac{1-t}{2[t(1-t)]^{3/2}}x_0\mathrm{d}t$$

$$\Longleftrightarrow \frac{1}{\sqrt{s(1-s)}}x_s - \frac{1}{\sqrt{t(1-t)}}x_t = \left(\frac{s}{\sqrt{s(1-s)}} - \frac{t}{\sqrt{t(1-t)}}\right)x_1 + \left(\frac{1-s}{\sqrt{s(1-s)}} - \frac{1-t}{\sqrt{t(1-t)}}\right)x_0$$

$$\Longleftrightarrow x_s = \frac{\sqrt{s(1-s)}}{\sqrt{t(1-t)}}x_t + \left(s - \frac{\sqrt{s(1-s)}}{\sqrt{t(1-t)}}t\right)x_1 + \left(1 - s - \frac{\sqrt{s(1-s)}}{\sqrt{t(1-t)}}(1-t)\right)x_0.$$

When $x_t \sim \mathcal{N}((1-t)x_0 + tx_1, t(1-t))$, we have:

$$x_s = \frac{\sqrt{s(1-s)}}{\sqrt{t(1-t)}}\left((1-t)x_0 + tx_1 + \sqrt{t(1-t)}\epsilon\right) + \left(s - \frac{\sqrt{s(1-s)}}{\sqrt{t(1-t)}}t\right)x_1$$

$$+ \left(1 - s - \frac{\sqrt{s(1-s)}}{\sqrt{t(1-t)}}(1-t)\right)x_0$$

$$= (1-s)x_0 + sx_1 + \sqrt{s(1-s)}\epsilon.$$

Hence, once the singularity is skipped by a stochastic step, following the PF-ODE reversely will preserve the marginals in this case. $\square$

## B.3 Derivation of the CBT Objective

Given $(\boldsymbol{x}, \boldsymbol{y}) \sim q_{\text{data}}(\boldsymbol{x}, \boldsymbol{y})$, $\boldsymbol{x}_t \sim q_{t|0T}(\boldsymbol{x}_t|\boldsymbol{x}_0 = \boldsymbol{x}, \boldsymbol{x}_T = \boldsymbol{y})$ and an estimate of $\hat{\boldsymbol{x}}_r = \hat{\boldsymbol{x}}_\phi(\boldsymbol{x}_t, t, \boldsymbol{y})$ based on the pre-trained score predictor $\boldsymbol{s}_\phi$ with the first-order ODE solver in Eqn. (16), our goal is to derive the alternative estimation of $\hat{\boldsymbol{x}}_r = \hat{\boldsymbol{x}}(\boldsymbol{x}_t, t, r, \boldsymbol{x}, \boldsymbol{y}) = a_r \boldsymbol{y} + b_r \boldsymbol{x} + c_r \boldsymbol{z}$ used in CBT, where $\boldsymbol{z} = \frac{\boldsymbol{x}_t - a_t \boldsymbol{y} - b_t \boldsymbol{y}}{c_t} \sim \mathcal{N}(\mathbf{0}, \boldsymbol{I})$ and $a_r, b_r, c_r$ are defined in Eqn. (11). We begin with the estimator with pre-trained score model and first-order ODE solver:

$$\boldsymbol{x}_r = \frac{\alpha_r \rho_r \bar{\rho}_r}{\alpha_t \rho_t \bar{\rho}_t} \boldsymbol{x}_t + \frac{\alpha_r}{\rho_T^2} \left[ \left( \bar{\rho}_r^2 - \frac{\bar{\rho}_t \rho_r \bar{\rho}_r}{\rho_t} \right) \boldsymbol{x}_\phi(\boldsymbol{x}_t, t, \boldsymbol{y}) + \left( \rho_r^2 - \frac{\rho_t \rho_r \bar{\rho}_r}{\bar{\rho}_t} \right) \frac{\boldsymbol{y}}{\alpha_T} \right], \tag{32}$$

where $\boldsymbol{x}_\phi$ is the equivalent data predictor of the score predictor $\boldsymbol{s}_\phi$. By the transformation between data and score predictor $\boldsymbol{x}_\phi = \frac{\boldsymbol{x}_t - a_t \boldsymbol{x}_T + c_t^2 \boldsymbol{s}_\phi}{b_t}$ and substituting the score predictor $\boldsymbol{s}_\phi$ with the score estimator $\nabla_{\boldsymbol{x}_t} q_{t|0T}(\boldsymbol{x}_t|\boldsymbol{x}_0 = \boldsymbol{x}, \boldsymbol{x}_T = \boldsymbol{y})$, we have:

$$\boldsymbol{x}_r = \frac{\alpha_r \rho_r \bar{\rho}_r}{\alpha_t \rho_t \bar{\rho}_t} \boldsymbol{x}_t + \frac{\alpha_r}{\rho_T^2} \left[ \left( \bar{\rho}_r^2 - \frac{\bar{\rho}_t \rho_r \bar{\rho}_r}{\rho_t} \right) \boldsymbol{x} + \left( \rho_r^2 - \frac{\rho_t \rho_r \bar{\rho}_r}{\bar{\rho}_t} \right) \frac{\boldsymbol{y}}{\alpha_T} \right], \tag{33}$$

By expressing $\boldsymbol{x}_t = a_t \boldsymbol{y} + b_t \boldsymbol{x} + c_t \boldsymbol{z}$, we could derive the corresponding coefficients for $\boldsymbol{x}, \boldsymbol{y}, \boldsymbol{z}$ on the right-hand side.

For $\boldsymbol{y}$:

$$\frac{\alpha_r \rho_r \bar{\rho}_r}{\alpha_t \rho_t \bar{\rho}_t} a_t + \frac{\alpha_r}{\alpha_T \rho_T^2} \left( \rho_r^2 - \frac{\rho_t \rho_r \bar{\rho}_r}{\bar{\rho}_t} \right) = \frac{\alpha_r \rho_r \bar{\rho}_r}{\alpha_t \rho_t \bar{\rho}_t} \frac{\bar{\alpha}_t \bar{\rho}_t^2}{\rho_T^2} + \frac{\alpha_r}{\alpha_T \rho_T^2} \left( \rho_r^2 - \frac{\rho_t \rho_r \bar{\rho}_r}{\bar{\rho}_t} \right)$$

$$\stackrel{(i)}{=} \frac{\alpha_r \rho_r \bar{\rho}_r}{\alpha_T \bar{\rho}_t} \frac{\rho_t}{\rho_T^2} + \frac{\alpha_r}{\alpha_T \rho_T^2} \left( \rho_r^2 - \frac{\rho_t \rho_r \bar{\rho}_r}{\bar{\rho}_t} \right) = \frac{\bar{\alpha}_r \rho_r^2}{\rho_T^2} = a_r, \tag{34}$$

where $(i)$ is due to the fact $\bar{\alpha}_t = \frac{\alpha_t}{\alpha_T}$.

For $\boldsymbol{x}$:

$$\frac{\alpha_r \rho_r \bar{\rho}_r}{\alpha_t \rho_t \bar{\rho}_t} b_t + \frac{\alpha_r}{\rho_T^2} \left( \bar{\rho}_r^2 - \frac{\bar{\rho}_t \rho_r \bar{\rho}_r}{\rho_t} \right) = \frac{\alpha_r \rho_r \bar{\rho}_r}{\alpha_t \rho_t \bar{\rho}_t} \frac{\alpha_t \bar{\rho}_t^2}{\rho_T^2} + \frac{\alpha_r}{\rho_T^2} \left( \bar{\rho}_r^2 - \frac{\bar{\rho}_t \rho_r \bar{\rho}_r}{\rho_t} \right) = \frac{\alpha_r \bar{\rho}_r^2}{\rho_T^2} = b_r. \tag{35}$$

For $\boldsymbol{z}$:

$$\frac{\alpha_r \rho_r \bar{\rho}_r}{\alpha_t \rho_t \bar{\rho}_t} c_t = \frac{\alpha_r \rho_r \bar{\rho}_r}{\alpha_t \rho_t \bar{\rho}_t} \frac{\alpha_t \bar{\rho}_t \rho_t}{\rho_T} = \frac{\alpha_r \bar{\rho}_r \rho_r}{\rho_T} = c_r. \tag{36}$$

Hence, we have the alternative model-free estimator $\hat{\boldsymbol{x}}_r = \hat{\boldsymbol{x}}(\boldsymbol{x}_t, t, r, \boldsymbol{x}, \boldsymbol{y}) = a_r \boldsymbol{y} + b_r \boldsymbol{x} + c_r \boldsymbol{z}$, where $\boldsymbol{z} \sim \mathcal{N}(\mathbf{0}, \boldsymbol{I})$ is the same Gaussian noise used in sampling $\boldsymbol{x}_t = a_t \boldsymbol{y} + b_t \boldsymbol{x} + c_t \boldsymbol{z}$. Substituting $\hat{\boldsymbol{x}}_\phi(\boldsymbol{x}_t, t, r, \boldsymbol{y})$ in the CBD objective in Eqn. (17) with $\hat{\boldsymbol{x}}(\boldsymbol{x}_t, t, r, \boldsymbol{x}, \boldsymbol{y})$ gives the CBT objective in Eqn. (20).

## B.4 Network Parameterization

First, we show the detailed network parameterization for DDBM in Table. 1. Denote the neural network as $\boldsymbol{F}_\theta$, the data predictor $\boldsymbol{x}_\theta(\boldsymbol{x}, t, \boldsymbol{y})$ is given by:

$$\boldsymbol{x}_\theta(\boldsymbol{x}_t, t, \boldsymbol{y}) = c_{\text{skip}}(t) \boldsymbol{x}_t + c_{\text{out}}(t) \boldsymbol{F}_\theta(c_{\text{in}}(t) \boldsymbol{x}_t, c_{\text{noise}}(t), \boldsymbol{y}), \tag{37}$$

where

$$c_{\text{in}}(t) = \frac{1}{\sqrt{a_t^2 \sigma_T^2 + b_t^2 \sigma_0^2 + 2 a_t b_t \sigma_{0T} + c_t}}, \quad c_{\text{out}}(t) = \sqrt{a_t^2 (\sigma_T^2 \sigma_0^2 - \sigma_{0T}^2) + \sigma_0^2 c_t} c_{\text{in}}(t),$$

$$c_{\text{skip}}(t) = (b_t \sigma_0^2 + a_t \sigma_{0T}) c_{\text{in}}^2(t), \quad c_{\text{noise}}(t) = \frac{1}{4} \log t. \tag{38}$$

and

$$a_t = \frac{\bar{\alpha}_t \rho_t^2}{\rho_T^2}, \quad b_t = \frac{\alpha_t \bar{\rho}_t^2}{\rho_T^2}, \quad c_t = \frac{\alpha_t^2 \bar{\rho}_t^2 \rho_t^2}{\rho_T^2}, \quad \sigma_0^2 = \text{Var}[\boldsymbol{x}_0], \sigma_T^2 = \text{Var}[\boldsymbol{x}_T], \sigma_{0T} = \text{Cov}[\boldsymbol{x}_0, \boldsymbol{x}_T]. \tag{39}$$

It can be verified that, with the variable substitution $\tilde{t} = t - \epsilon$, we have $a_{\tilde{\epsilon}} = 0, b_{\tilde{\epsilon}} = 1, c_{\tilde{\epsilon}} = 0$ and thus have $c_{\text{skip}}(\tilde{\epsilon}) = 1$ and $c_{\text{out}}(\tilde{\epsilon}) = 0$.

Meanwhile, we could generally parameterize the data predictor $\boldsymbol{x_\theta}$ with the one-step first-order solver from $t$ to $\epsilon$, i.e.:

$$\boldsymbol{f_\theta}(\boldsymbol{x}_t, t, \boldsymbol{y}) = \frac{\alpha_\epsilon \rho_\epsilon \bar{\rho}_\epsilon}{\alpha_t \rho_t \bar{\rho}_t} \boldsymbol{x}_t + \frac{\alpha_\epsilon}{\rho_T^2} \left[ \left( \bar{\rho}_\epsilon^2 - \frac{\bar{\rho}_t \rho_\epsilon \bar{\rho}_\epsilon}{\rho_t} \right) \boldsymbol{x}_\theta(\boldsymbol{x}_t, t, \boldsymbol{y}) + \left( \rho_\epsilon^2 - \frac{\rho_t \rho_\epsilon \bar{\rho}_\epsilon}{\bar{\rho}_t} \right) \frac{\boldsymbol{y}}{\alpha_T} \right], \quad (40)$$

which naturally satisfies $f(\boldsymbol{x}_\epsilon, \epsilon, \boldsymbol{y}) = \boldsymbol{x}_\epsilon$.

## B.5 Asymptotic Analysis of CBD

**Proposition 3.2.** *Given $\Delta t_{\max} = \max_t \{t - r(t)\}$ and let $\boldsymbol{h_\phi}(\cdot, \cdot, \cdot)$ be the consistency function of the empirical bridge ODE taking the form in Eqn. (8). Assume $\boldsymbol{h_\theta}$ is a Lipschitz function, i.e., there exists $L > 0$, such that for all $t \in [\epsilon, T - \gamma], \boldsymbol{x}_1, \boldsymbol{x}_2, \boldsymbol{y}$, we have $\|\boldsymbol{h_\theta}(\boldsymbol{x}_1, t, \boldsymbol{y}) - \boldsymbol{h_\theta}(\boldsymbol{x}_2, t, \boldsymbol{y})\|_2 \leq L\|\boldsymbol{x}_1 - \boldsymbol{x}_2\|_2$. Meanwhile, assume that for all $t, r \in [\epsilon, T - \gamma], \boldsymbol{y} \sim q_{\text{data}}(\boldsymbol{y}) := \mathbb{E}_{\boldsymbol{x}}[q_{\text{data}}(\boldsymbol{x}, \boldsymbol{y})]$, the ODE solver $\hat{\boldsymbol{x}}_\phi(\cdot, t, r, \boldsymbol{y})$ has local error uniformly bounded by $O((t - r)^{p+1})$ with $p \geq 1$. Then, if $\mathcal{L}_{\text{CBD}}^{\Delta t_{\max}} = 0$, we have: $\sup_{t, \boldsymbol{x}, \boldsymbol{y}} \|\boldsymbol{h_\theta}(\boldsymbol{x}, t, \boldsymbol{y}) - \boldsymbol{h_\phi}(\boldsymbol{x}, t, \boldsymbol{y})\|_2 = O((\Delta t_{\max})^p)$.*

Most of the proof directly follows the original consistency models analysis [58], with minor differences in the discrete timestep intervals (i.e., non-overlapped in [58] and overlapped in ours) and the form of marginal distribution between $p_t(\boldsymbol{x}_t)$ for the diffusion ODE and $q_{t|T}(\boldsymbol{x}_t|\boldsymbol{x}_T = \boldsymbol{y})$ for the bridge ODE.

*Proof.* Given $\mathcal{L}_{\text{CBD}}^{\Delta t_{\max}} = 0$, we have:

$$\mathbb{E}_{q_{\text{data}}(\boldsymbol{x}, \boldsymbol{y})q_{t|0T}(\boldsymbol{x}_t|\boldsymbol{x}_0=\boldsymbol{x}, \boldsymbol{x}_T=\boldsymbol{y})} \mathbb{E}_{t,r} \left[ \lambda(t) d \left( \boldsymbol{h_\theta}(\boldsymbol{x}_t, t, \boldsymbol{y}) - \boldsymbol{h_{\theta^-}}(\hat{\boldsymbol{x}}_\phi(\boldsymbol{x}_t, t, r, \boldsymbol{y}), r, \boldsymbol{y}) \right) \right] = 0 \quad (41)$$

Since $\lambda(t) > 0$, and for $t \in [\epsilon, T - \gamma]$, $q_{t|0T}(\boldsymbol{x}_t|\boldsymbol{x}_0 = \boldsymbol{x}, \boldsymbol{y}_0 = \boldsymbol{y})$ takes the form of $\mathcal{N}(a_t\boldsymbol{x}_T + b_t\boldsymbol{x}_0, c_t\boldsymbol{I})$ with $c_t > 0$, which entails for any $\boldsymbol{x}_t, t \in [\epsilon, T - \gamma]$, $q_{t|T}(\boldsymbol{x}_t|\boldsymbol{x}_T = \boldsymbol{y}) = \mathbb{E}_{\boldsymbol{x}}[q_{t|0T}(\boldsymbol{x}_t|\boldsymbol{x}_0 = \boldsymbol{x}, \boldsymbol{x}_T = \boldsymbol{y})] > 0$. Hence, Eqn. (41) implies that for all $t \in [\epsilon, T - \gamma], (\boldsymbol{x}, \boldsymbol{y}) \sim q_{\text{data}}(\boldsymbol{x}, \boldsymbol{y}), \boldsymbol{x}_t \sim q_{t|0T}(\boldsymbol{x}_t|\boldsymbol{x}_0 = \boldsymbol{x}, \boldsymbol{x}_T = \boldsymbol{y})$, we have:

$$d \left( \boldsymbol{h_\theta}(\boldsymbol{x}_t, t, \boldsymbol{y}) - \boldsymbol{h_{\theta^-}}(\hat{\boldsymbol{x}}_\phi(\boldsymbol{x}_t, t, r(t), \boldsymbol{y}), r(t), \boldsymbol{y}) \right) \equiv 0, \quad (42)$$

By the nature of the distance metric function $d$ and the stopgrad operator, we then have:

$$\boldsymbol{h_\theta}(\boldsymbol{x}_t, t, \boldsymbol{y}) \equiv \boldsymbol{h_{\theta^-}}(\hat{\boldsymbol{x}}_\phi(\boldsymbol{x}_t, t, r(t), \boldsymbol{y}), r(t), \boldsymbol{y}) \equiv \boldsymbol{h_\theta}(\hat{\boldsymbol{x}}_\phi(\boldsymbol{x}_t, t, r(t), \boldsymbol{y}), r(t), \boldsymbol{y}). \quad (43)$$

Define the error term at timestep $t \in [\epsilon, T - \gamma]$ as:

$$\boldsymbol{e}_t := \boldsymbol{h_\theta}(\boldsymbol{x}_t, t, \boldsymbol{y}) - \boldsymbol{h_\phi}(\boldsymbol{x}_t, t, \boldsymbol{y}). \quad (44)$$

We have:

$$\begin{aligned}
\boldsymbol{e}_t &= \boldsymbol{h_\theta}(\boldsymbol{x}_t, t, \boldsymbol{y}) - \boldsymbol{h_\phi}(\boldsymbol{x}_t, t, \boldsymbol{y}) \\
&= \boldsymbol{h_\theta}(\hat{\boldsymbol{x}}_\phi(\boldsymbol{x}_t, t, r(t), \boldsymbol{y}), r(t), \boldsymbol{y}) - \boldsymbol{h_\phi}(\boldsymbol{x}_{r(t)}, r(t), \boldsymbol{y}) \\
&= \boldsymbol{h_\theta}(\hat{\boldsymbol{x}}_\phi(\boldsymbol{x}_t, t, r(t), \boldsymbol{y}), r(t), \boldsymbol{y}) - \boldsymbol{h_\theta}(\boldsymbol{x}_{r(t)}, r(t), \boldsymbol{y}) \\
&\quad + \boldsymbol{h_\theta}(\boldsymbol{x}_{r(t)}, r(t), \boldsymbol{y}) - \boldsymbol{h_\phi}(\boldsymbol{x}_{r(t)}, r(t), \boldsymbol{y}) \\
&= \boldsymbol{h_\theta}(\hat{\boldsymbol{x}}_\phi(\boldsymbol{x}_t, t, r(t), \boldsymbol{y}), r(t), \boldsymbol{y}) - \boldsymbol{h_\theta}(\boldsymbol{x}_{r(t)}, r(t), \boldsymbol{y}) + \boldsymbol{e}_{r(t)}.
\end{aligned}$$

Since $\boldsymbol{h_\theta}$ is Lipschitz with constant $L$ and the ODE solver $\hat{\boldsymbol{x}}_\phi(\cdot, t, r, \boldsymbol{y})$ is bounded by $O((t - r)^{p+1})$ with $p \geq 1$, we have:

$$\begin{aligned}
\|\boldsymbol{e}_t\|_2 &\leq \|\boldsymbol{e}_{r(t)}\|_2 + L\|\hat{\boldsymbol{x}}_\phi(\boldsymbol{x}_t, t, r(t), \boldsymbol{y}) - \boldsymbol{x}_{r(t)}\|_2 \\
&= \|\boldsymbol{e}_{r(t)}\|_2 + L \cdot O((t - r(t))^{p+1}) \\
&= \|\boldsymbol{e}_{r(t)}\|_2 + O((t - r(t))^{p+1}).
\end{aligned}$$

From the boundary condition of the consistency function, we have:

$$\boldsymbol{e}_\epsilon = \boldsymbol{h_\theta}(\boldsymbol{x}_\epsilon, \epsilon, \boldsymbol{y}) - \boldsymbol{h_\phi}(\boldsymbol{x}_\epsilon, \epsilon, \boldsymbol{y}) = \boldsymbol{x}_\epsilon - \boldsymbol{x}_\epsilon = \boldsymbol{0}.$$

Denote $r_m(t)$ as applying $r$ on $t$ for $m$ times, since $\Delta t_{\min} = \min_t\{t - r(t)\}$ exists, there exists $N$ such that $r_n(t) = \epsilon$ for $n \geq N$. We thus have:

$$\|\boldsymbol{e}_t\|_2 \leq \|\boldsymbol{e}_\epsilon\|_2 + \sum_{k=1}^{N} O((r_{k-1}(t) - r_k(t))^{p+1})$$

$$= \sum_{k=1}^{N} O((r_{k-1}(t) - r_k(t))^{p+1})$$

$$= \sum_{k=1}^{N} (r_{k-1}(t) - r_k(t)) O((r_{k-1}(t) - r_k(t))^{p})$$

$$\leq \sum_{k=1}^{N} (r_{k-1}(t) - r_k(t)) O((\Delta t_{\max})^{p})$$

$$= O((\Delta t_{\max})^{p}) \sum_{k=1}^{N} (r_{k-1}(t) - r_k(t))$$

$$= O((\Delta t_{\max})^{p})(t - \epsilon)$$

$$\leq O((\Delta t_{\max})^{p})(T - \epsilon)$$

$$= O((\Delta t_{\max})^{p}).$$

$\square$

### B.6 Connection between CBD & CBT

**Proposition 3.3.** *Given $\Delta t_{\max} = \max_t\{t - r(t)\}$ and assume $d, \boldsymbol{h_\theta}, f, g$ are twice continuously differentiable with bounded second derivatives, the weighting function $\lambda(\cdot)$ is bounded, and $\mathbb{E}[\|\nabla_{\boldsymbol{x}_t} \log q_{t|T}(\boldsymbol{x}_t|\boldsymbol{x}_T)\|_2^2] < \infty$. Meanwhile, assume that $\mathcal{L}_{\mathrm{CBD}}^{\Delta t_{\max}}$ employs the one-step ODE solver in Eqn. (16) with ground truth pre-trained score model, i.e., $\forall t \in [\epsilon, T - \gamma], \boldsymbol{y} \sim q_{\mathrm{data}}(\boldsymbol{y}): \boldsymbol{s_\phi}(\boldsymbol{x}_t, t, \boldsymbol{y}) \equiv \nabla_{\boldsymbol{x}_t} \log q_{t|T}(\boldsymbol{x}_t|\boldsymbol{x}_T = \boldsymbol{y})$. Then, we have: $\mathcal{L}_{\mathrm{CBD}}^{\Delta t_{\max}} = \mathcal{L}_{\mathrm{CBT}}^{\Delta t_{\max}} + o(\Delta t_{\max})$.*

The core technique for building the connection between consistency distillation and consistency training with Taylor Expansion also directly follows [58]. The major difference lies in the form of the bridge ODE and the general noise schedule & the first-order ODE solver studied in our work.

*Proof.* First, for a twice continuously differentiable, multivariate, vector-valued function $\boldsymbol{h}(\boldsymbol{x}, t, \boldsymbol{y})$, denote $\partial_k \boldsymbol{h}(\boldsymbol{x}, t, \boldsymbol{y})$ as the Jacobian of $\boldsymbol{h}$ over the $k$-th variable. Consider the CBD objective with first-order ODE solver in Eqn. (16) (ignore terms taking expectation for notation simplicity):

$$\mathcal{L}_{\mathrm{CBD}}^{\Delta t_{\max}} = \mathbb{E}\left[\lambda(t) d\left(\boldsymbol{h_\theta}(\boldsymbol{x}_t, t, \boldsymbol{y}), \boldsymbol{h_{\theta^-}}(k_1(t, r)\boldsymbol{x}_t + k_2(t, r)\boldsymbol{x_\phi} + k_3(t, r)\boldsymbol{y}, r, \boldsymbol{y})\right)\right], \quad (45)$$

where $k_1(t, r) = \frac{\alpha_r \rho_r \bar\rho_r}{\alpha_t \rho_t \bar\rho_t}, k_2(t, r) = \frac{\alpha_r}{\rho_T^2}\left(\bar\rho_r^2 - \frac{\bar\rho_t \rho_r \bar\rho_r}{\rho_t}\right), k_3(t, r) = \frac{\alpha_r}{\alpha_T \rho_T^2}\left(\rho_r^2 - \frac{\rho_t \rho_r \bar\rho_r}{\bar\rho_t}\right)$ are coefficients of $\boldsymbol{x}_t, \boldsymbol{x_\phi}, \boldsymbol{y}$ in the first-order ODE solver in Eqn. (16), $\boldsymbol{x_\phi}$ is pre-trained data predictor. By applying first-order Taylor expansion on Eqn. (45), we have:

$$\mathcal{L}_{\mathrm{CBD}}^{\Delta t_{\max}}$$
$$=\mathbb{E}\left[\lambda(t) d\left(\boldsymbol{h_\theta}(\boldsymbol{x}_t, t, \boldsymbol{y}), \boldsymbol{h_{\theta^-}}(\boldsymbol{x}_t + (k_1(t, r) - 1)\boldsymbol{x}_t + k_2(t, r)\boldsymbol{x_\phi} + k_3(t, r)\boldsymbol{y}, t + (r - t), \boldsymbol{y})\right)\right]$$
$$=\mathbb{E}\left[\lambda(t) d\left(\boldsymbol{h_\theta}(\boldsymbol{x}_t, t, \boldsymbol{y}), \boldsymbol{h_{\theta^-}}(\boldsymbol{x}_t, t, \boldsymbol{y}) + \partial_1 \boldsymbol{h_{\theta^-}}(\boldsymbol{x}_t, t, \boldsymbol{y})[(k_1(t, r) - 1)\boldsymbol{x}_t + k_2(t, r)\boldsymbol{x_\phi} + k_3(t, r)\boldsymbol{y}]\right.\right.$$
$$\left.\left. + \partial_2 \boldsymbol{h_{\theta^-}}(\boldsymbol{x}_t, t, \boldsymbol{y})(r - t) + o(|t - r|)\right)\right].$$

Here the error term *w.r.t.* the first variable can be obtained by applying Taylor expansion on $k(t, r) = k(t, t) + \partial_2 k(t, t)(r - t) + o(|t - r|)$ with $k_1(t, t) - 1 = 0, k_2(t, t) = k_3(t, t) = 0$. By applying

Taylor expansion on $d$, we have:

$$
\begin{aligned}
&\mathcal{L}_{\mathrm{CBD}}^{\Delta t_{\max}}\\
=&\mathbb{E}\{\lambda(t)d(\boldsymbol{h}_{\boldsymbol{\theta}}(\boldsymbol{x}_t,t,\boldsymbol{y}),\boldsymbol{h}_{\boldsymbol{\theta}^-}(\boldsymbol{x}_t,t,\boldsymbol{y}))+\lambda(t)\partial_2 d(\boldsymbol{h}_{\boldsymbol{\theta}}(\boldsymbol{x}_t,t,\boldsymbol{y}),\boldsymbol{h}_{\boldsymbol{\theta}^-}(\boldsymbol{x}_t,t,\boldsymbol{y}))[\\
&\quad \partial_1 \boldsymbol{h}_{\boldsymbol{\theta}^-}(\boldsymbol{x}_t,t,\boldsymbol{y})[(k_1(t,r)-1)\boldsymbol{x}_t+k_2(t,r)\boldsymbol{x}_{\boldsymbol{\phi}}+k_3(t,r)\boldsymbol{y}]+\partial_2\boldsymbol{h}_{\boldsymbol{\theta}^-}(\boldsymbol{x}_t,t,\boldsymbol{y})(r-t)+o(|t-r|)]\}\\
=&\mathbb{E}\{\lambda(t)d(\boldsymbol{h}_{\boldsymbol{\theta}}(\boldsymbol{x}_t,t,\boldsymbol{y}),\boldsymbol{h}_{\boldsymbol{\theta}^-}(\boldsymbol{x}_t,t,\boldsymbol{y}))\}\\
&\quad +\mathbb{E}\{\lambda(t)\partial_2 d(\boldsymbol{h}_{\boldsymbol{\theta}}(\boldsymbol{x}_t,t,\boldsymbol{y}),\boldsymbol{h}_{\boldsymbol{\theta}^-}(\boldsymbol{x}_t,t,\boldsymbol{y}))[\partial_1\boldsymbol{h}_{\boldsymbol{\theta}^-}(\boldsymbol{x}_t,t,\boldsymbol{y})[(k_1(t,r)-1)\boldsymbol{x}_t+k_2(t,r)\boldsymbol{x}_{\boldsymbol{\phi}}+k_3(t,r)\boldsymbol{y}]]\}\\
&\quad +\mathbb{E}\{\lambda(t)\partial_2 d(\boldsymbol{h}_{\boldsymbol{\theta}}(\boldsymbol{x}_t,t,\boldsymbol{y}),\boldsymbol{h}_{\boldsymbol{\theta}^-}(\boldsymbol{x}_t,t,\boldsymbol{y}))\partial_2\boldsymbol{h}_{\boldsymbol{\theta}^-}(\boldsymbol{x}_t,t,\boldsymbol{y})(r-t)\}+\mathbb{E}\{o(|t-r|)\}.
\end{aligned}
$$

Then we focus on the term related to the first-order ODE solver:

$$(k_1(t,r)-1)\boldsymbol{x}_t+k_2(t,r)\boldsymbol{x}_{\boldsymbol{\phi}}+k_3(t,r)\boldsymbol{y}.$$

By the transformation between data and score predictor $\boldsymbol{x}_{\boldsymbol{\phi}}=\frac{\boldsymbol{x}_t-a_t\boldsymbol{x}_T+c_t^2\boldsymbol{s}_{\boldsymbol{\phi}}}{b_t}$, and substitute $\boldsymbol{s}_{\boldsymbol{\phi}}(\boldsymbol{x}_t,t,\boldsymbol{y})$ with $\nabla_{\boldsymbol{x}_t}\log q_{t|T}(\boldsymbol{x}_t|\boldsymbol{x}_T=\boldsymbol{y})$, we have:

$$(k_1(t,r)-1)\boldsymbol{x}_t+k_2(t,r)\frac{\boldsymbol{x}_t-a_t\boldsymbol{x}_T+c_t^2\nabla_{\boldsymbol{x}_t}\log q_{t|T}(\boldsymbol{x}_t|\boldsymbol{x}_T=\boldsymbol{y})}{b_t}+k_3(t,r)\boldsymbol{y}.$$

Next, substituting the score $\nabla_{\boldsymbol{x}_t}\log q_{t|T}(\boldsymbol{x}_t|\boldsymbol{x}_T=\boldsymbol{y})$ with the unbiased estimator:

$$\mathbb{E}[\nabla_{\boldsymbol{x}_t}\log q_{t|0T}(\boldsymbol{x}_t|\boldsymbol{x}_0,\boldsymbol{x}_T)|\boldsymbol{x}_t,\boldsymbol{x}_T=\boldsymbol{y}]=\mathbb{E}\left[-\frac{\boldsymbol{x}_t-(a_t\boldsymbol{x}_T+b_t\boldsymbol{x}_0)}{c_t^2}|\boldsymbol{x}_t,\boldsymbol{x}_T=\boldsymbol{y}\right]$$

We then have:

$$
\begin{aligned}
&\mathbb{E}\{\lambda(t)\partial_2 d(\boldsymbol{h}_{\boldsymbol{\theta}}(\boldsymbol{x}_t,t,\boldsymbol{y}),\boldsymbol{h}_{\boldsymbol{\theta}^-}(\boldsymbol{x}_t,t,\boldsymbol{y}))[\partial_1\boldsymbol{h}_{\boldsymbol{\theta}^-}(\boldsymbol{x}_t,t,\boldsymbol{y})[(k_1(t,r)-1)\boldsymbol{x}_t+k_2(t,r)\boldsymbol{x}_{\boldsymbol{\phi}}+k_3(t,r)\boldsymbol{y}]]\}\\
=&\mathbb{E}\{\lambda(t)\partial_2 d(\boldsymbol{h}_{\boldsymbol{\theta}}(\boldsymbol{x}_t,t,\boldsymbol{y}),\boldsymbol{h}_{\boldsymbol{\theta}^-}(\boldsymbol{x}_t,t,\boldsymbol{y}))[\partial_1\boldsymbol{h}_{\boldsymbol{\theta}^-}(\boldsymbol{x}_t,t,\boldsymbol{y})[\\
&(k_1(t,r)-1)\boldsymbol{x}_t+k_2(t,r)\frac{\boldsymbol{x}_t-a_t\boldsymbol{x}_T+c_t^2\mathbb{E}\left[-\frac{\boldsymbol{x}_t-(a_t\boldsymbol{x}_T+b_t\boldsymbol{x}_0)}{c_t^2}|\boldsymbol{x}_t,\boldsymbol{x}_T=\boldsymbol{y}\right]}{b_t}+k_3(t,r)\boldsymbol{y}\\
\overset{(i)}{=}&\mathbb{E}\{\lambda(t)\partial_2 d(\boldsymbol{h}_{\boldsymbol{\theta}}(\boldsymbol{x}_t,t,\boldsymbol{y}),\boldsymbol{h}_{\boldsymbol{\theta}^-}(\boldsymbol{x}_t,t,\boldsymbol{y}))[\partial_1\boldsymbol{h}_{\boldsymbol{\theta}^-}(\boldsymbol{x}_t,t,\boldsymbol{y})[\\
&(k_1(t,r)-1)\boldsymbol{x}_t+k_2(t,r)\frac{\boldsymbol{x}_t-a_t\boldsymbol{x}_T-c_t^2\frac{\boldsymbol{x}_t-(a_t\boldsymbol{x}_T+b_t\boldsymbol{x}_0)}{c_t^2}}{b_t}+k_3(t,r)\boldsymbol{y}]\\
=&\mathbb{E}\{\lambda(t)\partial_2 d(\boldsymbol{h}_{\boldsymbol{\theta}}(\boldsymbol{x}_t,t,\boldsymbol{y}),\boldsymbol{h}_{\boldsymbol{\theta}^-}(\boldsymbol{x}_t,t,\boldsymbol{y}))[\partial_1\boldsymbol{h}_{\boldsymbol{\theta}^-}(\boldsymbol{x}_t,t,\boldsymbol{y})[k_1(t,r)\boldsymbol{x}_t+k_2(t,r)\boldsymbol{x}+k_3(t,r)\boldsymbol{y}-\boldsymbol{x}_t],
\end{aligned}
$$

where $(i)$ comes from the law of total expectation. Then we apply Taylor expansion in the reverse direction:

$$
\begin{aligned}
&\mathcal{L}_{\mathrm{CBD}}^{\Delta t_{\max}}\\
=&\mathbb{E}\{\lambda(t)d(\boldsymbol{h}_{\boldsymbol{\theta}}(\boldsymbol{x}_t,t,\boldsymbol{y}),\boldsymbol{h}_{\boldsymbol{\theta}^-}(\boldsymbol{x}_t,t,\boldsymbol{y}))\}\\
&\quad +\mathbb{E}\{\lambda(t)\partial_2 d(\boldsymbol{h}_{\boldsymbol{\theta}}(\boldsymbol{x}_t,t,\boldsymbol{y}),\boldsymbol{h}_{\boldsymbol{\theta}^-}(\boldsymbol{x}_t,t,\boldsymbol{y}))[\partial_1\boldsymbol{h}_{\boldsymbol{\theta}^-}(\boldsymbol{x}_t,t,\boldsymbol{y})[k_1(t,r)\boldsymbol{x}_t+k_2(t,r)\boldsymbol{x}+k_3(t,r)\boldsymbol{y}-\boldsymbol{x}_t]]\}\\
&\quad +\mathbb{E}\{\lambda(t)\partial_2 d(\boldsymbol{h}_{\boldsymbol{\theta}}(\boldsymbol{x}_t,t,\boldsymbol{y}),\boldsymbol{h}_{\boldsymbol{\theta}^-}(\boldsymbol{x}_t,t,\boldsymbol{y}))\partial_2\boldsymbol{h}_{\boldsymbol{\theta}^-}(\boldsymbol{x}_t,t,\boldsymbol{y})(r-t)\}+\mathbb{E}\{o(|t-r|)\}.\\
=&\mathbb{E}\{\lambda(t)[d(\boldsymbol{h}_{\boldsymbol{\theta}}(\boldsymbol{x}_t,t,\boldsymbol{y}),\boldsymbol{h}_{\boldsymbol{\theta}^-}(\boldsymbol{x}_t,t,\boldsymbol{y}))\\
&\quad +\partial_2 d(\boldsymbol{h}_{\boldsymbol{\theta}}(\boldsymbol{x}_t,t,\boldsymbol{y}),\boldsymbol{h}_{\boldsymbol{\theta}^-}(\boldsymbol{x}_t,t,\boldsymbol{y}))[\partial_1\boldsymbol{h}_{\boldsymbol{\theta}^-}(\boldsymbol{x}_t,t,\boldsymbol{y})[k_1(t,r)\boldsymbol{x}_t+k_2(t,r)\boldsymbol{x}+k_3(t,r)\boldsymbol{y}-\boldsymbol{x}_t]]\\
&\quad +\partial_2 d(\boldsymbol{h}_{\boldsymbol{\theta}}(\boldsymbol{x}_t,t,\boldsymbol{y}),\boldsymbol{h}_{\boldsymbol{\theta}^-}(\boldsymbol{x}_t,t,\boldsymbol{y}))\partial_2\boldsymbol{h}_{\boldsymbol{\theta}^-}(\boldsymbol{x}_t,t,\boldsymbol{y})(r-t)]\}+\mathbb{E}\{o(|t-r|)\}\\
=&\mathbb{E}\{\lambda(t)[d(\boldsymbol{h}_{\boldsymbol{\theta}}(\boldsymbol{x}_t,t,\boldsymbol{y}),\boldsymbol{h}_{\boldsymbol{\theta}^-}(\boldsymbol{x}_t,t,\boldsymbol{y})+\partial_1\boldsymbol{h}_{\boldsymbol{\theta}^-}(\boldsymbol{x}_t,t,\boldsymbol{y})[k_1(t,r)\boldsymbol{x}_t+k_2(t,r)\boldsymbol{x}+k_3(t,r)\boldsymbol{y}-\boldsymbol{x}_t]\\
&\quad +\partial_2\boldsymbol{h}_{\boldsymbol{\theta}^-}(\boldsymbol{x}_t,t,\boldsymbol{y})(r-t))]\}+\mathbb{E}\{o(|t-r|)\}\\
=&\mathbb{E}\{\lambda(t)[d(\boldsymbol{h}_{\boldsymbol{\theta}}(\boldsymbol{x}_t,t,\boldsymbol{y}),\boldsymbol{h}_{\boldsymbol{\theta}^-}(k_1(t,r)\boldsymbol{x}_t+k_2(t,r)\boldsymbol{x}+k_3(t,r)\boldsymbol{y},r,\boldsymbol{y}))]\}+\mathbb{E}\{o(|t-r|)\}\\
\overset{(ii)}{=}&\mathbb{E}\{\lambda(t)[d(\boldsymbol{h}_{\boldsymbol{\theta}}(a_t\boldsymbol{y}+b_t\boldsymbol{x}+c_t\boldsymbol{z},t,\boldsymbol{y}),\boldsymbol{h}_{\boldsymbol{\theta}^-}(a_r\boldsymbol{y}+b_r\boldsymbol{x}+c_r\boldsymbol{z},r,\boldsymbol{y}))]\}+o(|t-r|)\\
=&\mathcal{L}_{\mathrm{CBT}}^{\Delta t_{\max}}+o(|t-r|),
\end{aligned}
$$

where $(ii)$ follows the derivation in Eqn. (33) – Eqn. (36), and $\boldsymbol{z}\sim\mathcal{N}(\boldsymbol{0},\boldsymbol{I})$. $\qquad\square$

## C  Additional Experimental Details

### C.1  Details of Training and Sampling Configurations

We train CDBMs based on a series of pre-trained DDBMs. For two image-to-image translation tasks, we directly use the pre-trained checkpoints provided by DDBM's [72] official repository.[2] For image inpainting, we re-train a model with the same I$^2$SB style noise schedule, network parameterization, and timestep scheme in Table. 1, as well as the overall network architecture. Unlike the training setup in I$^2$SB, our network is conditioned on $x_T = y$ following DDBM and takes the class information of ImageNet as input, which we refer to as the base DDBM model for image inpainting on ImageNet. The model is initialized with the class-conditional version on ImageNet $256 \times 256$ of guided diffusion [10]. We used a global batch size of 256 and a constant learning rate of `1e-5` with mixed precision (fp16) to train the model for 200k steps. We train the model with 8 NVIDIA A800 80G GPUs for 9.5 days, achieving the FID reported in Table. 3 with the first-order ODE solver in Eqn. (16).

For training CDBMs, we use a global batch size of 128 and a learning rate of `1e-5` with mixed precision (fp16) for all datasets using 8 NVIDIA A800 80G GPUs. For the constant training schedule $r(t) = t - \Delta t$, we train the model for 50k steps, while for the sigmoid-style training schedule, we train the model for $6s$ steps, e.g., 30k or 60k steps, due to numerical instability when $t - r(t)$ is small. For CBD, training a model for 50k steps on a dataset with $256 \times 256$ resolution takes ${\sim}2.5$ days, while CBT takes ${\sim}1.5$ days. In this work, we normalize all images within $[-1, 1]$ and adopt the RAdam [27, 34] optimizer.

For sampling, we use a uniform timestep for all baselines with the ODE solver and CDBM on two image-to-image translation tasks with $\epsilon = 0.0001, T = 1.0$. For CDBM on image inpainting on ImageNet, we manually assign the second timestep to $T - 0.1$ and make other timesteps uniformly distributed between $[\epsilon, T - 0.1)$, which we find yields better empirical performance on this task.

### C.2  Details of Training Schedule for CDBM

We illustrate the effect of the hyperparamter $b$ in the sigmoid-like training schedule $r(t) = t(1 - \frac{1}{q^{\lfloor \text{iters}/s \rfloor}})(1 + \frac{k}{1+e^{bt}})$. Note that we further manually enforce $r(t)$ to satisfy $\Delta t_{\max}$ and $\Delta t_{\min}$.

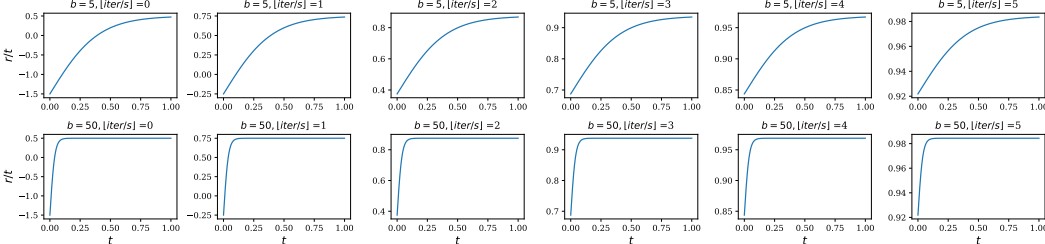

Figure 6: Illustration of the effect of the parameter $b$ on the sigmoid-style training schedule.

### C.3  License

We list the used datasets, codes, and their licenses in Table 4.

Table 4: The used datasets, codes and their licenses.

| Name | URL | Citation | License |
|---|---|---|---|
| Edges→Handbags | https://github.com/junyanz/pytorch-CycleGAN-and-pix2pix | [23] | BSD |
| DIODE-Outdoor | https://diode-dataset.org/ | [62] | MIT |
| ImageNet | https://www.image-net.org | [9] | \ |
| Guided-Diffusion | https://github.com/openai/guided-diffusion | [10] | MIT |
| I$^2$SB | https://github.com/NVlabs/I2SB | [33] | CC-BY-NC-SA-4.0 |
| DDBM | https://github.com/alexzhou907/DDBM | [72] | \ |

---

[2]`https://github.com/alexzhou907/DDBM`

# D  Additional Samples

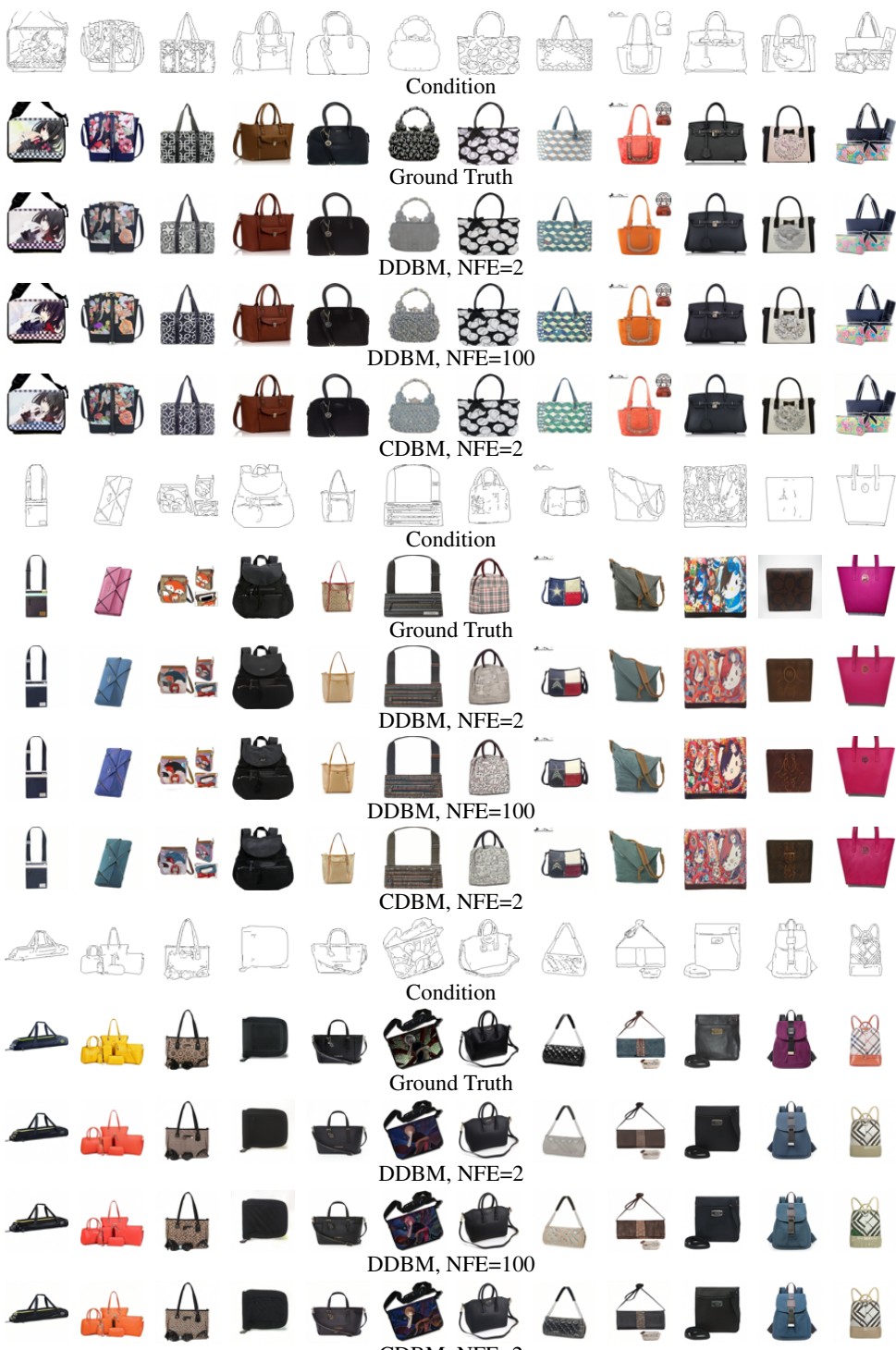

Figure 7: Additional Samples for Edges → Handbags.

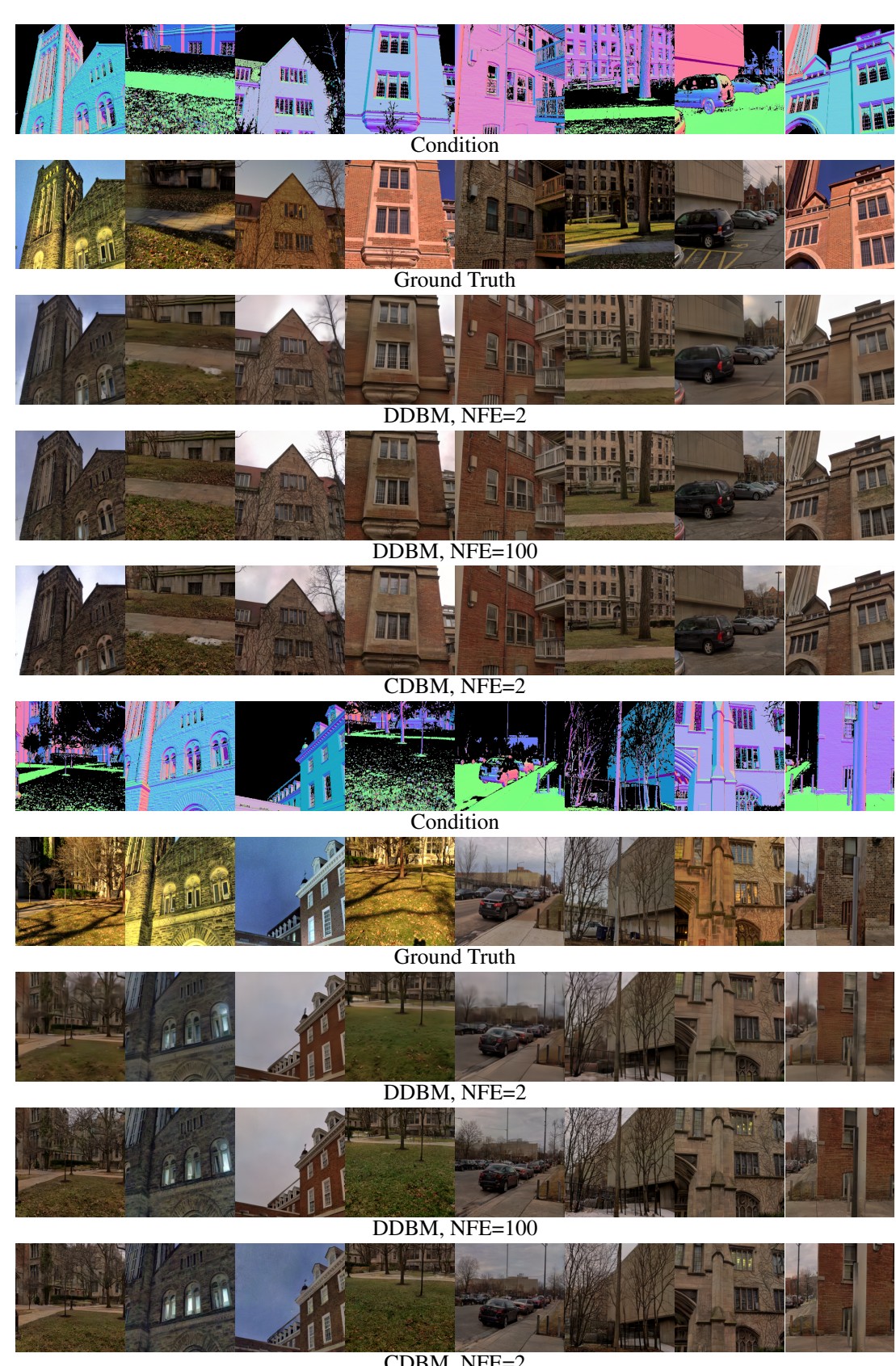

Figure 8: Additional Samples for DIODE-Outdoor.

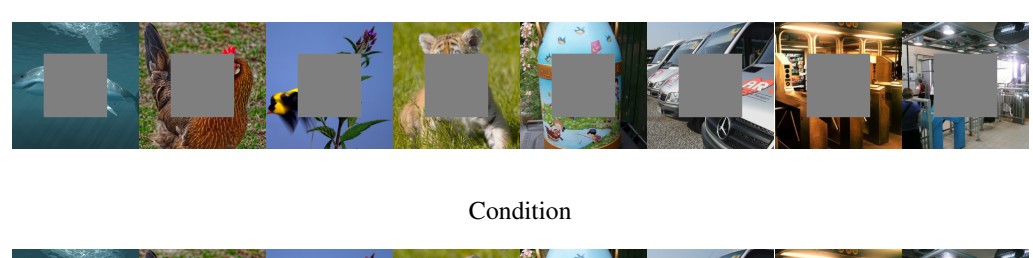

Condition

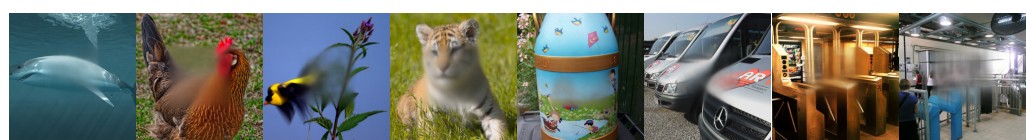

Ground Truth

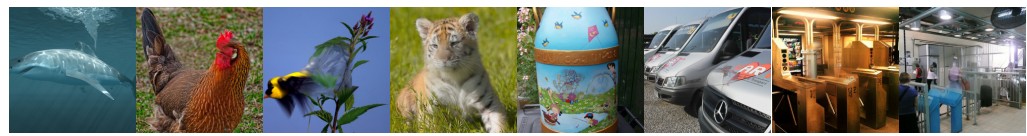

DDBM, NFE=2

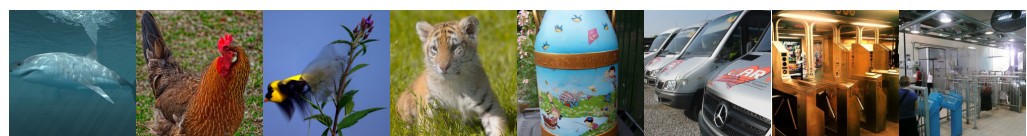

DDBM, NFE=8

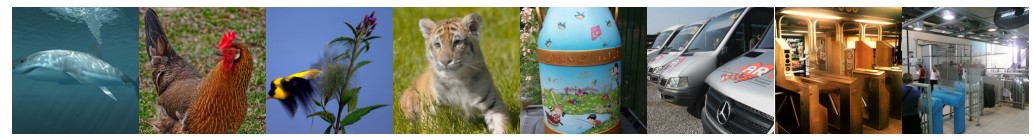

CDBM, NFE=2

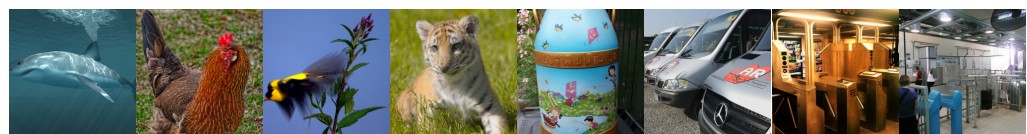

DDBM, NFE=10

CDBM, NFE=10

Figure 9: Additional Samples for ImageNet $256 \times 256$.

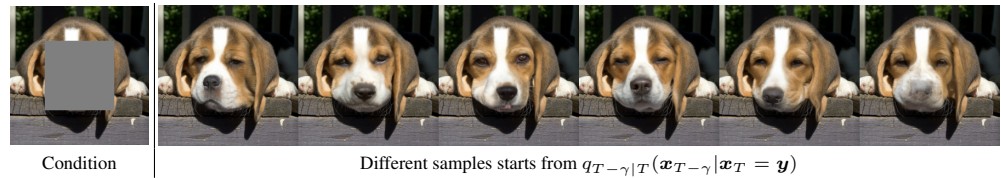

Condition                    Different samples starts from $q_{T-\gamma|T}(\boldsymbol{x}_{T-\gamma}|\boldsymbol{x}_T = \boldsymbol{y})$

Figure 10: Demonstration of sample diversity of the deterministic ODE sampler.

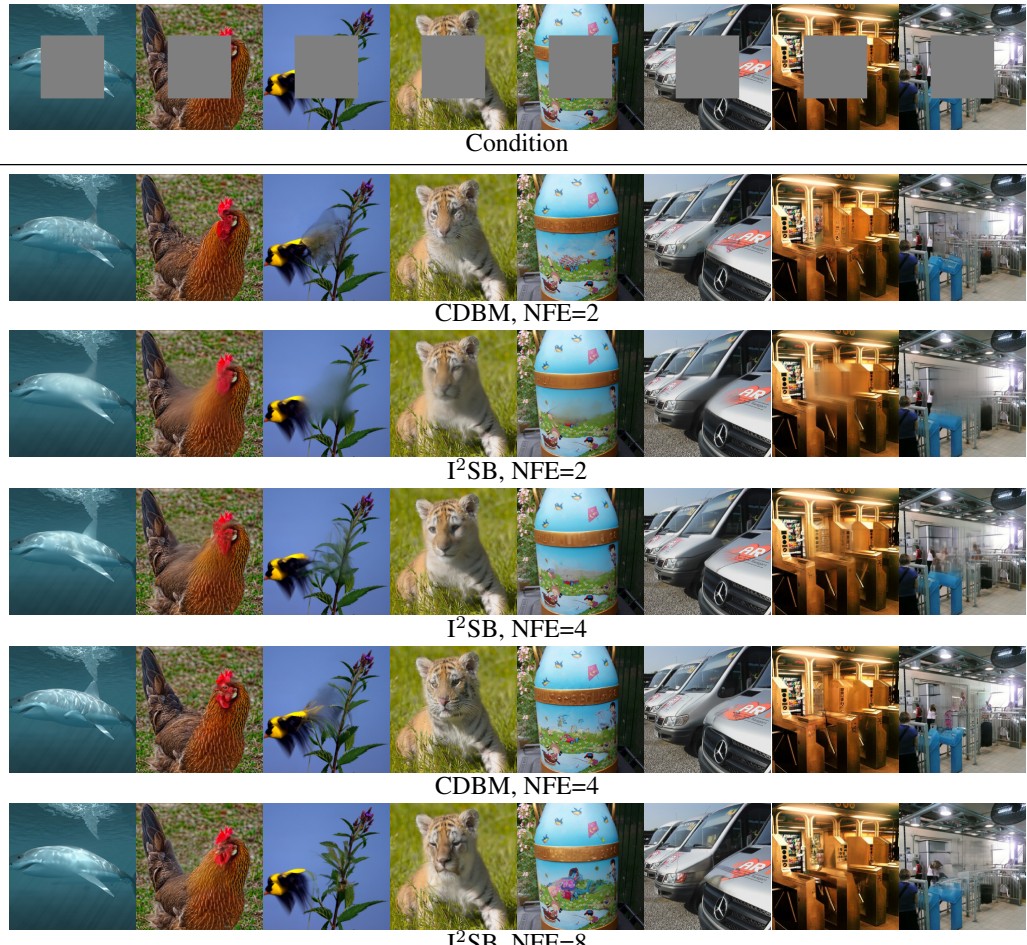

Figure 11: Qualitative comparison between CDBM and I$^2$SB baseline on ImageNet $256 \times 256$. Note that here the base model of CDBM is different from the officially released checkpoint of I$^2$SB we used for evaluation.

