# OpenReview forum: "Consistency Diffusion Bridge Models"
_NeurIPS.cc/2024/Conference — NeurIPS 2024 poster_

### Official Review · Reviewer_nr1H · 2024-07-07

**Soundness:** 3
**Presentation:** 3
**Contribution:** 3
**Rating:** 7
**Confidence:** 4

**Summary:**

This paper works on efficient sampling of the denoising diffusion bridge models (DDBMs) or similar. In particular, the paper proposes to extend the consistency models (CMs) to DDBMs.

CMs are generative models developed in the context of improving the sampling cost of diffusion-based generative models (DBGMs). Note that sampling of the probability flows of the DBGMs runs the integrators up to a few hundred steps or more. Similar to common DBGMs, assume we have a forward diffusion whose initial distribution—data—is denoted by $x_0$. In particular, when we rewrite the forward diffusion in the probability flow (PF) form, we get an ODE whose marginal is equivalent to the original forward diffusion. We further assume that we have the ODE-integrators that run $t$-decreasing direction starting from any given $t \in (0, T]$ to $0$. Then, the CMs train to predict the output of the integrators, similar to denoising autoencoder, but the perturbation here is deterministic. Training such models uses the consistency condition. Once we train the CMs, we can use them instead of running a few hundred steps of DGBMs.

DDBMs are input-conditional bridge models. More specifically, assume we have two distributions and their joint distribution, e.g., one is an image distribution, and the other is its edge images. The bridge models learn a diffusion to connect one distribution to the other. Unlike common bridge models, however, DDBMs conditions on its initial point, e.g., an edge image, so that the path generated by the learned diffusion preserves the information of the initial value. For several applications, such as image-to-image translations, these DDBMs show several benefits, unlike common Markovian models; in particular, the initial coupling of the joint distributions is preferred. Nevertheless, DDBMs are also required to run the SDE-integrators (or ODE-integrators for its PFs) a few hundred steps, similar to any other diffusion-based models.

In order to overcome sampling inefficiency in DDBMs, the paper proposes to extend CMs to DDBMs.

Finally, the authors demonstrate the proposed method's effectiveness via various experiments, including image-to-image translations and image inpainting.

---------------------------------------
Updated the rating from 4 to 7 after the authors' rebuttal

**Strengths:**

One of the paper's key contributions is the introduction of a novel solution to improve the sampling of the DDBM-like models. The proposed method addresses one of the significant challenges in diffusion-based models and offers a fresh perspective on how state-of-the-art techniques can be extended to the DDBM-like models. Furthermore, the proposed method exhibits comparable performance and scalability compared to popular diffusion-based generative models with a fewer number of evaluations, further underscoring the practical effectiveness of the proposed method.

**Weaknesses:**

The authors tackle a highly interesting problem and reach out for an excellent solution. However, I find that the paper requires a major revision due to a few theoretical loopholes in extending the consistency models (CMs) to the denoising diffusion bridge models (DDBMs).

For example, in expressing CM in the form of a stochastic differential equation (SDE), the paper uses a definition that is not mathematically well-defined. Particularly, in Equation 2, the notation of a reverse-time standard Wiener process is mentioned. However, a reverse-time standard Wiener process is not defined in general; more precisely, it is tricky to define such a concept. The reason is that the definition of the It\'o integral is highly sensitive to the direction of time. Consequently, the change of variables used in calculus cannot be applied to SDE directly. As a result, we need to define an SDE-specific chain rule called Ito's lemma. Accordingly, one cannot rewrite an SDE by changing the sign of time. In order to apply the chain rule of calculus, the SDE literature uses the Stieltjes integral instead of the It\'o integral, and a corresponding SDE notation needs to be defined. If this approach were taken for the submitted paper, however, the denoising diffusion and consistency model would need to be defined accordingly.

In conclusion, this paper's development deviates from the fundamental assumptions of the It\'o integral-based SDEs. Note that the original CM circumvents this issue by using only ordinary differential equations (or partial differential equations). Therefore, unlike the It\'o integral, it is free from concerns about the time directions. However, this does not apply to SDEs.

In my understanding, one would be able to reach the same conclusion as the submitted paper while sticking to the fundamentals of SDEs (related to It'o\' diffusion). However, such modification would require a major revision from the current submission.

In addition, a few statements need to be addressed. In particular, to get an unbiased estimation of the expectation in Equation 15, one needs to sample $x_t$ and $x_T$ first and then sample $x_0$. However, the authors state that we can achieve this by sampling $x_0$ and $x_T$ first and then $x_t$. Such an estimator is biased except when we use a single sample.

**Questions:**

N/A

---

> ### Author Rebuttal · Authors · 2024-08-06
>
> Thank you for your supportive comments and valuable feedback. We would like to address the weaknesses as follows:
>
> > W1: This paper's development deviates from the fundamental assumptions of the It'o integral-based SDEs.
>
> **A**: We respectfully disagree with this statement and would like to address that **we are not expressing CM in the form of a stochastic differential equation (SDE).** The reverse SDE for denoising diffusion and DDBM in Eqn. (2) and Eqn. (7) are only mentioned once in Section 2 using the definition from established literature for completeness (which *can be removed without affecting the definition and development of CM* if it might cause a presentation issue). The consistency model for denoising diffusion and DDBM is defined with the PF-ODE in Eqn. (3) and Eqn. (8), respectively, which do not depend on the reverse SDE in Eqn. (2) and Eqn. (7).
>
> > W2: a few statements need to be addressed. In particular, to get an unbiased estimation of the expectation in Equation 15, one needs to sample $x_t$ and $x_T$ first and then sample $x_0$. However, the authors state that we can achieve this by sampling $x_0$ and $x_T$ first and then $x_t$. Such an estimator is biased except when we use a single sample.
>
> **A**: We appreciate the reviewer for rigorously pointing this out. We will revise this to: With a single sample $(x, y) \sim q_{\mathrm{data}}(x, y)$ and $x_t \sim q_{t|0T}(x_t|x_0=x, x_T=y)$, the score $\nabla_{x_t}\log q_{t|T}(x_t|x_T=y)$ can be estimated with $\nabla_{x_t}\log q_{t|0T}(x_t|x_0=x, x_T=y)$.

---

### Official Review · Reviewer_7nJA · 2024-07-11

**Soundness:** 3
**Presentation:** 4
**Contribution:** 2
**Rating:** 6
**Confidence:** 4

**Summary:**

In this paper, the authors combine Denoising Diffusion Bridge Models (DDBMs), that build a transport map between two arbitrary target distributions (that can be coupled) through a stochastic process, with recent advances on consistency techniques [1], that were originally designed for denoising diffusion models. The motivation herein is to speed up the inference process of DDBMs, that turns out to be costly due to a high number of neural network evaluations. Following the approach from [1], the authors propose to learn the *consistency function* of the probability flow associated to an arbitrary DDBM, which outputs a prediction of the target distribution given any input of the DDBM process $x_t$, for any time $t$. To do so, they derive a consistency loss, which can be seen as the natural extension of the loss of [1] to the bridge setting. Then, they propose two paradigms for this approach: either *consistency bridge distillation*, which assumes to have access to a pretrained DDBM model, and *consistency bridge training*, where no model is available. Their formulation encompasses popular designs of diffusion bridges such as the Brownian brigde, the Image-to-Image Schrödinger bridge [2], the original bridges designed for DDBMs [3]... The authors provide practical guidelines on the neural network parameterization and preconditioning, the design of the ODE solver and the consistency schedule. Finally, they conduct numerical experiments on high-dimensional datasets that demonstrate that the proposed consistency bridge model is faster than the original model while having equal or better generative performance. They notably find out that fine-tuning a pretrained DDBM with consistency training provides better performance under lower computational budget than distillation.

[1] Consistency models. Song et al. 2023.

[2] I2SB: Image-to-Image Schrödinger Bridge. Liu et al. 2023.

[3] Denoising diffusion bridge models. Zhou et al. 2023

**Strengths:**

- Although this paper adapts lots of elements from [1] to the bridge setting, which may amortize the novelty of this contribution, I think that this contribution may be impactful for practical usage as it encompasses a large formulation of DDBMs and demonstrates great empirical performance, see Section 4.
- The paper is well written and easy to follow. In particular, the mathematical statements are clear to understand.
- The presentation of the background is well done and helps the readability of the paper.
- The comparison with related work in the numerics section is well conducted.
- All details on the experiments are available in the paper, which is a really good point for reproducibility.

[1] Consistency models. Song et al. 2023.

**Weaknesses:**

- The current paper does not exhibit any particular novelty compared to the setting from [1], but this is not a major weakness to me.
- I think that the paper does not provide enough intuition on the design of a good consistency schedule.

[1] Consistency models. Song et al. 2023.

**Questions:**

- Could you give further details and intuition on the design of the consistency schedules $r(t)$, which seem to be crucial so that consistency works (even if these reasons are heuristic) ? In particular, could you give examples of schedules that may be adapted in theory but do not work in practice ? It still unclear for me which design should be the best depending on the type of consistency (either distillation or training).
- I have a question on the approximation made on the score in Equation (15) for consistency training, which can actually also be addressed to standard consistency models. Theoretically, up to the terminal conditioning in the bridge setting, the score evaluated in $x$ at time $t$ has to be understood as a conditional expectation over the posterior distribution of $X_0$ given $X_t=x$ by Tweedie's formula. This distribution has a very specific behaviour: (i) when $t\approx0$ (close to the target distribution), it is concentrated like a Dirac mass; (ii), on the other hand when $t\approx T$ (full noise), the posterior is approximately equal to the target distribution itself. This is well highlighted in the concept of duality of log-concavity in [1]. Therefore, the approximation of the score as presented in the current paper (that is evaluating the integrand in only one point $x_0$) is relevant when $t\approx0$, but completely false when $t\approx T$. Do you have an estimation of the error made in the latter case for tractable cases ? Does it have an impact on the training for this part of the process ? Have you come up with an idea to correct this intrinsic bias ?

[1] Stochastic localization via iterative posterior sampling. Grenioux et al. 2024

**Limitations:**

The limitations are clearly indicated in Section 5. The authors notably acknowledge that bridge consistency models suffer from numerical instability (same as classic consistency models).

---

> ### Author Rebuttal · Authors · 2024-08-06
>
> Thank you for acknowledging our contribution and the valuable comments on our work. We would like to provide our responses as follows (in reverse order for better logical flow):
>
> > Q1: The approximation of the score as presented in the current paper (that is evaluating the integrand in only one point $x_0$) is relevant when $t \approx 0$, but completely false when $t \approx T$. Do you have an estimation of the error made in the latter case for tractable cases? Does it have an impact on the training for this part of the process? Have you come up with an idea to correct this intrinsic bias?
>
> **A**: Thank you for the question. We think that the original CM paper [1] has largely addressed the problem of justifying the use of the mentioned data score for consistency training. Specifically, in Theorem 6 of the CM paper, they reached a conclusion that, when the interval between two timesteps for training CM becomes infinitely small, then the gradient of the consistency training loss and the consistency distillation loss with the true score under Euler ODE solver will be identical. Therefore, the mismatch between the data score (posterior) and true score (marginal) can be addressed by shrinking the interval between the two timesteps in training CM. In practice, this becomes a principle for designing a consistency schedule that ensures $t - r(t)$ to be as small as possible, at least at the end of training. Nevertheless, we admit the inaccurate estimation will have an impact and we think one way to address it is by proposing more fine-grained consistency schedules, which is an interesting and important research direction.
>
>
> > Q2: Could you give further details and intuition on the design of the consistency schedules $r(t)$, which seem to be crucial so that consistency works (even if these reasons are heuristic)? In particular, could you give examples of schedules that may be adapted in theory but do not work in practice? It still unclear to me which design should be the best depending on the type of consistency (either distillation or training).
>
> **A**: Thank you for the question. First, the design space of training consistency model, such as consistency schedules, loss weighting, and distance metric in consistency loss, has become a specialized area of research, and there are several papers (e.g., [2, 3]) target on investigating better designs for these components. In this work, we mainly adopt the design guideline proposed in [3] and verified its effectiveness in a different setting, i.e., the training of a CM for the PF-ODE of DDBMs. In specific, the consistency schedule should gradually shrink $t - r(t)$ to be small to avoid optimization problems and error accumulation due to such small time intervals as much as possible, which is intuitively elaborated in Sec 3.2 in [3]. In our experiments, we found that the gradually shrinking schedule generally yields better performance than a fixed interval and could be applied to both consistency distillation and training. Hence, our conclusions on the consistency schedule are consistent with the intuition and insights of recent works studying this on diffusion models. However, how to rationalize the design of the process shrinking $t - r(t)$ (e.g., in conjunction with the accuracy of the score estimation discussed in Q1) is still an active area of research.
>
> [1] Song, Yang, et al. "Consistency Models." International Conference on Machine Learning. ICML, 2023.
>
> [2] Song, Yang, and Prafulla Dhariwal. "Improved techniques for training consistency models." ICLR 2024.
>
> [3] Geng, Zhengyang, et al. "Consistency Models Made Easy." arXiv preprint 2024.

---

> > ### Comment · Reviewer_7nJA · 2024-08-11
> > **Answer to the rebuttal**
> >
> > I would like to thank the authors for their precise answers. I will keep my score unchanged.

---

### Official Review · Reviewer_2JER · 2024-07-14

**Soundness:** 3
**Presentation:** 3
**Contribution:** 2
**Rating:** 7
**Confidence:** 3

**Summary:**

Motivated by faster sampling time, the authors investigate consistency model techniques applied to denoising diffusion bridge models, a particular formulation of conditioned diffusion model whereby the forward process is a mixture of diffusion bridge SDEs, with a conditioned terminal point.

In particular, the authors adapt typical consistency model training regimed known as "consistency model training" - training the model from scratch -  and "consistency model distillation" - distilling from a pretrained diffusion (bridge) model.

**Strengths:**

As far as I am aware consistency methods have not been applied to diffusion bridge models, though the significance of this is unclear vs performing consistency training on diffusion models or flow matching models.

The empirical performance showcased by the author's implementation appears competetitive to the baselines the authors consider.

**Weaknesses:**

Denoising diffusion bridge models are a particular formulation of conditioned diffusion model whereby the forward process is given by the SDE of a diffusion bridge, with a conditioned terminal point. This has the benefit of data to data translation, but in my opinion the authors exaggerate the significance of this and I would not consider it "a new family of generative models" [line 4]. Given the consistency methods applied to diffusion models transfer trivially to diffusion bridge models, I am of the opinion that the novelty of the methodological contribution is quite limited.

Furthermore I would also question the usefulness of data to data generative models which are deterministic. For any ill-posed inverse problem, practitioners would want to be able to sample from many possible uncorrupted sample given a single corrupted sample, including some example provided here such as for inpainting.

For consistency models or indeed any flow type model, a Gaussian is typically chosen as marginal which adds stochasticity and permits the PF ODE to match the diffusion model in marginal distributions. This is not the case for diffusion bridge models with conditioned terminal point, there is no longer any stochasticity and hence the ODE derived cannot match the marginals of the stochastic generative backward process. Hence I wonder if PF-ODE of a diffusion bridge model even makes sense in the same way as a diffusion model? Given all the consistency approaches rely on the deterministic component, I believe this is quite fundamental and needs to be addressed.

There are a number of related works that could be discussed. In particular Augmented bridge matching (https://arxiv.org/abs/2311.06978) which draws an equivalence between denoising diffusion bridge models and conditioned bridge matching.


Other:
>  tractable class of Schrödinger Bridge and simulation-free, non-iterative training procedure [32, 6]

[32,6] consider regular bridge matching without any connection to an optimal coupling and hence although tractable just a bridge matching model and not a schrodinger bridge. This is a common mistake.

**Questions:**

See weaknesses.

**Limitations:**

See weaknesses.

---

> ### Author Rebuttal · Authors · 2024-08-06
>
> Thank you for your valuable comments and feedback on our work. We would like to provide our responses as follows:
>
> > W1: The significance of DDBM is exaggerated as a new family of generative models
>
> **A**: Thank you for pointing this out and we agree that DDBM itself can not be called as a family of generative models. We will revise this to "a new formulation of generative modeling".
>
>
> > W2: Given the consistency methods applied to diffusion models transfer trivially to diffusion bridge models, I am of the opinion that the novelty of the methodological contribution is quite limited
>
> **A**: While the high-level idea and core techniques of utilizing CM to accelerate the sampling of DDBM might appear direct, we would like to emphasize this work still makes some valid technical contributions to the community.
>
> First, we demonstrate the effectiveness and advantages of the presented first-order ODE sampler for DDBM, which could achieve competitive, or even superior performance to strong baselines with a deterministic sampling trajectory. To the best of our knowledge, there is no other better candidate in previous works. For example, DDBM [1] finds that using the ODE sampler in EDM [2] yielded poor performance and proposed a hybrid sampler that alternates between ODE and SDE steps. And I$^2$SB [3] uses a stochastic posterior sampling scheme. Neither of them can naturally fit the CM paradigm.
>
> Second, our presented unified framework of empirical design space (i.e., noise schedule, network parameterization, and preconditioning) can encompass a wide range of diffusion bridges, which is completely decoupled from their different theoretical premises (e.g., either the method belongs to bridge matching or conditioned bridge matching). As a result, people could efficiently reuse the successful empirical designs of previous diffusion bridges in a unified way according to our framework. For example, in our paper, we retrain an image inpainting model with DDBM's mathematical formulation and I$^2$SB's noise schedule, network parameterization, and preconditioning that has comparable performance to the original I$^2$SB. And we can further build a consistency model on top of it in a unified manner.
>
> > W3: I would also question the usefulness of data to data generative models which are deterministic. For any ill-posed inverse problem, practitioners would want to be able to sample from many possible uncorrupted sample given a single corrupted sample, including some example provided here such as for inpainting.
>
> **A**: We would like to argue that the deterministic sampling process of data to data generative models, such as the presented first-order ODE sampler for DDBM, does not necessarily produce a deterministic sample given a fixed input. For the inverse problem, **it does produce diverse uncorrupted samples given a single corrupted sample.** This is because of the fact that the PF-ODE of DDBM is only well-defined for $0 \le t < T$. Thus, the valid way to simulate the PF-ODE is to start from $q_{T-\gamma|T}(x_{T-\gamma}|x_T)$ for some $\gamma > 0$, which introduces stochasticity and will result in sample diversity given a fixed $x_T$. In practice, once we have a plausible approximation of initial distribution, the ODE sampler will work similarly to its counterpart in noise-to-data generative models. We also provide a concrete illustration of the sample diversity with our first-order ODE sampler in the image inpainting task on the one-page pdf in the common response.
>
> On the other hand, deterministic samplers may also have other advantages such as improved sampling efficiency (while ensuring sample diversity). For example, in our replicated image inpainting model with DDBM's formulation, the first-order ODE sampler is notably better than a first-order SDE sampler when NFE is small (e.g., $<100$). Another piece of evidence is that I$^2$SB's officially released checkpoint also adopts a deterministic sampler that removes the stochasticity of posterior sampling.
>
>
>
> > W4: For diffusion bridge models with conditioned terminal point, there is no longer any stochasticity and hence the ODE derived cannot match the marginals of the stochastic generative backward process
>
>
> **A**: As we explained in W3 and addressed in DDBM [1], the PF-ODE derived by DDBM is well-defined and models the evolution of $q_{t|T}(x_t | x_T)$ when $0 \le t < T$. The singularity caused by the fixed terminal point can (and should) be removed by using a stochastic initial distribution, and doing so will enable valid ODE sampling and consistency modeling. We will add some notes about this point in our revised version. Furthermore, we find that the consistency losses in the current version allow us to sample $t=T$ from $\mathcal U(\epsilon, T)$ and we will revise the timestep distribution to $\mathcal U(\epsilon, T - \gamma)$ with a pre-specified $\gamma > 0$.
>
> > W5: There are a number of related works that could be discussed. In particular Augmented bridge matching, which draws an equivalence between denoising diffusion bridge models and conditioned bridge matching.
>
> **A**: Thank you for your suggestion. Augmented bridge matching is an excellent work that reveals and discusses how conditioning on the terminal point changes the theoretical properties of bridge matching, notably pointing out that conditioned bridge matching (or DDBM) preserves the empirical coupling. We have already mentioned this work in our initial submission and will add more discussion about it and the bridge matching technique in our revised version.
>
> > W6: [32,6] consider regular bridge matching without any connection to an optimal coupling and hence although tractable just a bridge matching model and not a schrodinger bridge. This is a common mistake.
>
> **A**: We thank the reviewer for pointing this out. We will add the discussion about bridge matching and properly classify these works as bridge matching & conditional bridge matching procedures in the related works section.

---

> > ### Author Response · Authors · 2024-08-07
> > **References**
> >
> > [1] Zhou, Linqi, et al. "Denoising Diffusion Bridge Models." ICLR 2024.
> >
> > [2] Karras, Tero, et al. "Elucidating the design space of diffusion-based generative models." NeurIPS 2022.
> >
> > [3] Liu, Guan-Horng, et al. "I$^2$SB: Image-to-Image Schrödinger Bridge." ICML 2023.

---

> > ### Comment · Reviewer_2JER · 2024-08-11
> >
> > Thank you for the response.
> >
> > My primary concern over the validity of the PF-ODE is a theoretical concern not an implementation/ stability issue over the singularity at marginal points. Your response does not address my concern.
> >
> > My understanding is that the initial stochastic sampling step is only at sampling / post training. Does this have any theoretical basis?
> >
> > Let us ignore this heuristic and consider the actual reverse diffusion bridge between two points and the "ODE" the authors describe. The fact remains that this ODE is fully deterministic given the starting point as such it cannot match the marginals of the corresponding stochastic bridge it is determined from at all time intervals. Then it is not clear theoretically why the end marginal should match a data distribution?
> >
> > DDBM [1] does not derive the PF ODE and provides a very hand wavy reference to Song 2021, however the regime is different so again it is not clear whether the PF ODE holds. The Kolmogorov forward/ backward equations are for Markov processes and the conditioned bridge SDE is clearly not Markov.
> >
> > The same issue has been raised with DDBM in review: https://openreview.net/forum?id=FKksTayvGo by both a reviewer and public commenter. I am inclined to agree with the comments there.
> >
> >
> > > The dynamics of () makes it clear why it is not possible to define a probability-flow ODE matching the marginal distributions of () on : in this hypothetical ODE, i.e. (7), both the initial condition and the dynamics would be deterministic, with no randomness left. The mentioned "averaged out" results correspond to the evolution of this deterministic ODE.
> >
> > > ODE (7) is not a probability flow ODE matching the marginal distribution of SDE (6)
> >
> > This is quite a fundamental issue and without being addressed, I am leaning towards lowering my score.

---

> > > ### Author Response · Authors · 2024-08-11
> > > **Reply to Reviewer 2JER (Round 2)**
> > >
> > > We would like to make further clarification/discussion regarding the validity of the PF-ODE according to our understanding of DDBM.
> > >
> > > First of all, the singularity at marginal points of the ODE is not an implementation/stability issue. Instead, this ODE is theoretically well-defined only on $ 0 \le t < T $, which means it can not be simulated from the given starting point theoretically. The valid way of utilizing the ODE would be first simulating the SDE from $T$ to $T - \gamma$ for some $\gamma > 0$ and then following the ODE, thus the whole sampling process will be stochastic and matches the marginal of the SDE.
> > >
> > > Moreover, we think DDBM did give a sound derivation of the PF-ODE in Appendix A.3 of their paper, which utilizes the Kolmogorov forward/backward equations of the reference diffusion process rather than the diffusion bridge. According to this, we are inclined to agree with the reply of DDBM's authors regarding your mentioned two comments:
> > >
> > > > We stress that (and also updated the text to include this) that our proposed deterministic process is more technically involved than simply defining a deterministic initial condition and dynamics. In particular, the introduction of randomness comes from the fact that our ODE is only well-defined on $ 0 \le t < T $, as Doob’s h-function causes a singularity at the boundary $ T $. When sampling using the SDE, we need to approximate $ x_{T-\epsilon} \approx y $ and follow the backwards SDE. For the ODE, the source of randomness comes from the initial distribution $p(x_{T-\epsilon})$, which is not the same as $y$. Instead, we sample $x_{T-\epsilon}$ (specifically approximating $x_{T - \epsilon^{\prime}} \approx y$) and then taking an Euler-Maruyama back to $T - \epsilon$), with which we can then sample with the valid probability flow ODE. Note that this clears the singularity and injects randomness while enabling ODE sampling.
> > >
> > > In a summary, we think DDBM has already addressed the theoretical validity of the PF-ODE and we adopt their established conclusion as a basis of our work. However, we are open to further inspecting their conclusion if there are indeed critical/fundamental theoretical flaws to avoid propagating errors by follow-up works like ours.

---

> ### Comment · Reviewer_2JER · 2024-08-11
>
> It is not clear to me that the forward SDE and backward ODE proposed have the same marginals. Consider a brownian bridge x_t between 2 points x_0 and x_T. There is a non zero probability that x_t > L, say p(x_t>L|x_0, x_T)>l>0.  The heuristic proposed is supposedly valid for any gamma>0, then in the backward ODE process p(x_t>L|x_T) -> 0 as gamma -> 0 hence backward and forward do not coincide in distribution for all gamma > 0 ,so the argument proposed seems incorrect.
>
> The Kolmogorov forward equation typically requires the markov property which does not hold here. It is not clear to me if this is the issue or something else, I am aware of cases where Kolmogorov equations hold for non markov processes but this typically requires more justification than provided
>
>
> I would be happy to raise my score if this is cleared up but it seems lack of a valid probability flow ODE is a major flaw in the paper.

---

> > ### Author Response · Authors · 2024-08-12
> >
> > We are not totally clear about the Kolmogorov forward equation part you mentioned, but we believe the probability flow ODE part in DDBM is correct.
> >
> > Firstly, we would like to argue that **diffusion bridges are Markovian processes**, as the condition $x_T$ is **fixed ahead of time**. As a simple example, the Brownian bridge is a Markovian process. This is exercise 5.11 in Oksendal's book "Stochastic Differential Equations". In the forward process, given $s<u<t$, $x_t$ depends on $x_u$, $x_T$ and is independent of $x_s$. In the reverse process, $x_s$ depends on $x_u,x_T$ and is independent of $x_t$. This is intuitive since the diffusion bridge can be represented as a simple SDE and simulated towards a single direction.
> >
> > Secondly, we can **rigorously prove** that in the simple Brownian bridge case, simulating the SDE from $T$ to $T-\gamma$, and then simulating the ODE from $T-\gamma$ **maintains the marginals**. For simplification, assume $T=1$. In the time region $[1-\gamma,1]$, the SDE has no singularity so the marginal is the same as the forward process $p(x_t|x_0,x_1)=N(tx_1+(1-t)x_0,t(1-t))$ (you should agree with this). Then the ground-truth probability flow ODE can be derived as (we follow DDBM paper and omit the derivation process here, but we can provide the details if you doubt this)
> >
> > $$
> > \frac{dx_t}{dt}=\frac{1-2t}{2t(1-t)}x_t+\frac{1}{2(1-t)}x_1-\frac{1}{2t}x_0
> > $$
> > The analytic solution of the ODE from time $t$ to time $s<t$ is (also we omit the derivation, but we can provide the details if you doubt this)
> >
> > $$
> > \frac{x_s}{\sqrt{s(1-s)}}-\frac{x_t}{\sqrt{t(1-t)}}=\left(\frac{s}{\sqrt{s(1-s)}}-\frac{t}{\sqrt{t(1-t)}}\right)x_1+\left(\frac{1-s}{\sqrt{s(1-s)}}-\frac{1-t}{\sqrt{t(1-t)}}\right)x_0
> > $$
> > or
> > $$
> > x_s=\frac{\sqrt{s(1-s)}}{\sqrt{t(1-t)}}x_t+\left(s-\frac{\sqrt{s(1-s)}}{\sqrt{t(1-t)}}t\right)x_1+\left(1-s-\frac{\sqrt{s(1-s)}}{\sqrt{t(1-t)}}(1-t)\right)x_0
> > $$
> > When $x_t$ follows the marginal $N(tx_1+(1-t)x_0,t(1-t))$, it can be easily verified from the above relation that $x_s$ follows the marginal $N(sx_1+(1-s)x_0,s(1-s))$. Therefore, **in the simple case you mentioned, following the ODE reversely in time will maintain the marginals**.

---

> ### Comment · Reviewer_2JER · 2024-08-12
>
> I was meaning the reverse process is not Markovian but I do not think this is a problem.
>
>  Thank you for the clarification, I think you are correct - apologies! I appreciate it and this is very interesting and a good contribution to the paper. I think this proof should be included in the paper, unless I've missed it somewhere. The DDBM paper does not have this initial stochastic step so the second part of Theorem 1 of DDBM is incorrect.

---

> > ### Author Response · Authors · 2024-08-12
> >
> > We are glad to see your concerns are successfully addressed and would like to thank you for the valuable review and discussion. We agree that DDBM did not address the validity of the probability flow ODE properly in Theorem 1 of their paper and we also agree that this is quite important. We will add a discussion about this and the full proof under the simple Brownian bridge case in the revised paper.

---

### Official Review · Reviewer_NqhJ · 2024-07-20

**Soundness:** 3
**Presentation:** 3
**Contribution:** 2
**Rating:** 5
**Confidence:** 4

**Summary:**

This paper proposes to combine the consistency model (CM) with diffusion denoising bridge models (DDBMs) to build a consistency diffusion bridge model (CDBMs), which includes two paradigms of consistency bridge distillation and consistency bridge training. The experiments are conducted for image inpainting and image-to-image translation tasks.

**Strengths:**

+ This paper is well written and organized, with a clear structure for readers to follow up easily.

+ The unified view of design space of several different DDBM models is an interesting summary for a clear comparison over existing works.

+ The experiment's results especially with NFE=2 are promising to be comparable with other models with larger NFE steps as shown in Figure 4.

**Weaknesses:**

- The proposed method is reasonable but may not be very surprising. This work looks more like a direct combination of CM and DDBM, which models have been introduced a lot in previous works, including consistency distillation and consistency training.

- Question about the motivation to combine CM and DDBM. CM is designed to reduce NFE steps requirement for diffusion sampling. DDBM models can also be able to achieve this goal by learning transformation between two data distributions, such as what is claimed in I2SB, while sacrificing the advantage of unsupervised learning. What is the key motivation and necessity to combine these two together?

- In experiments results, some common metrics like PSNR and SSIM are not reported. In Table 3, the compared methods are most unsupervised learning approaches which are not trained with paired data. In this case, more NFE steps are expected. Therefore, for a fair comparison, the NFE for the compared baselines in Table 2 and 3 may also need to be noted.

- For qualitative results, the comparison with previous DDBM method under the same NFE would be appreciated, since some DDBM method such as I2SB and CDDB [1] also claimed enabling image generation or restoration within a very few NFE steps from 1-4. It would be important comparison to compare with these baselines under the same experiment settings including NFE.

- Due to the requirement for paired data training, the proposed model needs to be retrained to different tasks, which are limited compared with diffusion-based methods. What is the computational complexity and parameter generalization robustness to retrain the model on a new task?


[1] Chung, Hyungjin, Jeongsol Kim, and Jong Chul Ye. "Direct diffusion bridge using data consistency for inverse problems." Advances in Neural Information Processing Systems, 2023.

**Questions:**

Please see the weaknesses for specific questions.

**Limitations:**

The author discusses the limitations of the proposed method in the last section.

---

> ### Author Rebuttal · Authors · 2024-08-05
>
> Thank you for your valuable comments and feedback on our work. We provide our responses as follows:
>
> > W1: The proposed method may not be very surprising. This work looks more like a direct combination of CM and DDBM, which have been introduced in previous works.
>
> **A**: While the high-level idea of utilizing CM to accelerate the sampling of DDBM might appear direct, we would like to emphasize this work still makes some valid technical contributions to the community that can not be accomplished by simply combining the existing techniques from previous works.
>
> First, we demonstrate the effectiveness and advantages of the presented first-order ODE sampler for DDBM, which could achieve competitive, even superior performance to strong baselines with a deterministic sampling trajectory. To the best of our knowledge, there is no other better candidate in previous works. For example, DDBM [1] finds that using the ODE sampler in EDM [2] yielded poor performance and proposed a hybrid sampler that alternates between ODE and SDE steps. And I$^2$SB [3] uses a stochastic posterior sampling scheme. Neither of them can naturally fit the CM paradigm.
>
> Second, our presented unified framework of empirical design space (i.e., noise schedule, network parameterization, and preconditioning) can encompass a wide range of diffusion bridges, which is completely decoupled from their different theoretical premises. As a result, people could efficiently reuse the successful empirical designs of previous diffusion bridges in a unified way according to our framework. For example, in our paper, we retrain an image inpainting model with DDBM's mathematical formulation and I$^2$SB's noise schedule, network parameterization, and preconditioning that has comparable performance to the original I$^2$SB. And we can further build a consistency model on top of it in a unified manner.
>
>
> > W2: The key motivation and necessity to combine DDBM and CM is unclear.
>
> **A**: We thank the reviewer for this question. For DDBMs, the benefit of utilizing the paired data during training is not only reducing the NFE requirement for sampling but also further improving the saturated performance given numerous NFE steps over unsupervised diffusion models. For example, the DDBM baseline in [1] needs **hundreds** NFE steps to achieve the saturated performance in some image translation tasks, surpassing the saturated performance of diffusion models. We show that introducing CM to DDBM can achieve these SOTA results $\sim 50 \times$ faster. In our opinion, this is analogous to why we need CM on diffusion models. On the other hand, even though some diffusion bridges (e.g., I$^2$SB) can achieve decent performance in a few NFE steps (e.g, $\ge 10$ steps), we believe that further reducing $2.5 \times$ NFE steps still makes a valid contribution.
>
> > W3: Some common metrics like PSNR and SSIM are not reported in experimental results.
>
> **A**: Thank you for the suggestion. We will add these two metrics in the appendix of our revised paper. We show the results in the comment for a short overview.
>
> > W4: The NFE for the compared baselines in Table 2 and 3 may also need to be noted.
>
> **A**: Thank you for the suggestion, we will add the NFE of the baselines in the revised paper. Most results are directly taken from the DDBM [1] and I$^2$SB [3] and some of them only have a rough number of NFE steps. We put them in the comment below for a simple overview.
>
> > W5: For qualitative results, the comparison with previous DDBM method under the same NFE would be appreciated.
>
> **A**: We thank the reviewer for the suggestion. Regarding the performance under a few NFE of the I$^2$SB baseline, we want to address that the DDBM (ODE-1) baseline in Table 3 is our replicated version of I$^2$SB on the center image inpainting task. As demonstrated in the answer of W1, we train an image inpainting model with DDBM's mathematical formulation and I$^2$SB's noise schedule, network parameterization, and preconditioning that has comparable quantitative performance to the original I$^2$SB. We put the results of both models under the same NFE in the comment for a simple overview.
>
> As shown in the table, our replicated model can achieve the reported FID of 4.9 in I$^2$SB paper with 10 NFE steps. On the other hand, I$^2$SB only provides checkpoints with a fully deterministic forward process (i.e., the "OT-ODE" in their paper), which does not align with the DDBM formulation. Hence, we choose our version as the main baseline. We also add a qualitative comparison between I$^2$SB and CDBM in the one-page pdf.
>
> Besides, it seems like the mentioned CDDB [4] paper still needs $>50$, even thousands of NFE steps to achieve the saturated performance, which, again, justifies our motivation of introducing CM to achieve comparable performance with much fewer NFE steps.
>
>
> > W6: What is the computational complexity and parameter generalization robustness to retrain the model on a new task?
>
> **A**: The cost of the paired data training stage is mostly concentrated on the base DDBM model. For reference, training a DDBM from scratch on the DIODE (256 $\times$ 256) image translation task would need $\sim500$k iterations. In our experiment that replicates I$^2$SB, it takes us about 200k iterations to *fine-tune* a DDBM for center inpainting from an unsupervised diffusion model on ImageNet 256 $\times$ 256, which is far easier than learning the diffusion model on this dataset. Once we have the base DDBM model, the consistency distillation/fine-tuning typically takes $<100$k iterations using the data for training the DDBM, which is more computationally efficient compared to training the base DDBM model itself.
>
> For generalization robustness and efficiency on a new task, it would be an interesting research direction to figure out whether there is, or how to build a foundation model (such as a diffusion model) that can be efficiently transferred to a diffusion bridge on various new tasks by leveraging paired data.

---

> > ### Author Response · Authors · 2024-08-07
> > **Reference and Additional Results**
> >
> > ## Reference
> > [1] Zhou, Linqi, et al. "Denoising Diffusion Bridge Models." ICLR 2024.
> >
> > [2] Karras, Tero, et al. "Elucidating the design space of diffusion-based generative models." NeurIPS 2022.
> >
> > [3] Liu, Guan-Horng, et al. "I$^2$SB: Image-to-Image Schrödinger Bridge." ICML 2023.
> >
> > [4] Chung, Hyungjin, Jeongsol Kim, and Jong Chul Ye. "Direct diffusion bridge using data consistency for inverse problems." NeurIPS 2023.
> >
> > ---
> > ## Additional Results
> > * ### Results for W3 (PSNR & SSIM for image translation tasks):
> >
> > |                       | E2H             |                 | DIODE           |                 |
> > |-----------------------|-----------------|-----------------|-----------------|-----------------|
> > |                       | PSNR $\uparrow$ | SSIM $\uparrow$ | PSNR $\uparrow$ | SSIM $\uparrow$ |
> > | DDBM (heun, NFE=118)        |            8.36 |            0.29 |           22.64 |            0.63 |
> > | DDBM (ODE-1, NFE=2)   |           30.67 |            0.91 |           25.32 |            0.75 |
> > | DDBM (ODE-1, NFE=50)  |            29.1 |            0.89 |           23.52 |             0.70 |
> > | DDBM (ODE-1, NFE=100) |           28.91 |            0.89 |           23.41 |            0.69 |
> > | CBD (Ours, NFE=2)     |           25.25 |            0.84 |           22.68 |            0.65 |
> > | CBT (Ours, NFE=2)     |           27.97 |            0.88 |           23.44 |            0.68 |
> >
> > * ### Results for W4 (NFE of baselines in Table 2 & 3):
> >
> > For table 2:
> > | Method         | NFE      |
> > |----------------|----------|
> > | Pix2Pix        |        1 |
> > | DDIB           | $\ge 40$ |
> > | SDEdit         | $\ge 40$ |
> > | Rectified Flow | $\ge 40$ |
> > | I$^2$SB        | $\ge 40$ |
> >
> > For table 3:
> > | Method   | NFE  |
> > |----------|------|
> > | DDRM     |   20 |
> > | $\Pi$GDM |  100 |
> > | DDNM     |  100 |
> > | Palette  | 1000 |
> > | I$^2$SB  | > 10 |
> >
> >
> > * ### Results for W5 (Quantative results of the I$^2$SB baseline):
> > | Model                        | NFE=2 | NFE=3 | NFE=4 | NFE=5 | NFE=8 | NFE=10 |
> > |------------------------------|-------|-------|-------|-------|-------|--------|
> > | I$^2$SB (posterior sampling) | 12.49 |  8.55 |   7.10 |  6.38 |  5.49 |   5.26 |
> > | DDBM (ODE-1)                 | 17.17 | 11.17 |  8.21 |  6.77 |  5.18 |   4.81 |

---

> > > ### Comment · Reviewer_NqhJ · 2024-08-12
> > >
> > > Thanks for the author to answer my questions. Some of my concerns are addressed. (1) Note that the PSNR and SSIM results in results Table for W3 of DDBM (ODE-1, NFE=2) are largely higher than the proposed method CBD / CBT (NFE=2). Could the author explain more about the reason? (2) Regarding the response to W5, the key point is if the experiments comparison is fair, under same NFE. Thanks again for adding new compared results between proposed method and I2SB with the same NFE in the pdf file. It is kind of confusing about the argument with CDDB though. Although the CDDB paper also reports that more NFEs can help, the key is to compare in the same setting? The CDDB paper reports that "20 NFE CDDB outperforms 1000 NFE I2SB in PSNR by > 2 db". Can it suppose that CDDB would be a stronger baseline than I2SB to compare with?

---

> ### Author Response · Authors · 2024-08-13
>
> Thank you for your feedback and we would like to provide the response to your follow-up question as follows:
>
> > Q1: Note that the PSNR and SSIM results in Table for W3 of DDBM (ODE-1, NFE=2) are largely higher than the proposed method CBD / CBT (NFE=2). Could the author explain more about the reason?
>
> Thank you for the question. We would like to argue that, when evaluating generative models such as diffusion models that hold a certain level of sample diversity, the distortion metrics laid on the pixel space such as PSNR and SSIM may not be the proper major evaluator that aligns with sample quality. For example, DDBM (ODE-1, NFE=2) not only surpasses CBD / CBT (NFE=2) but also surpasses DDBM (ODE-1, NFE=100) in terms of PSNR & SSIM. One should agree that using hundreds of NFEs yields better sample quality than only two NFEs on the image translation task, though NFE=2 has better PSNR & SSIM than NFE=100, which can be justified by the blurry samples under NFE=2. Thus, following the community of diffusion models, we focus on FID as the major metric and claim the accelerated ratio according to this metric, where CBD / CBT (NFE=2) clearly surpasses DDBM (ODE-1, NFE=2). On the other hand, the results in Table for W3 actually further support our claim that CBD / CBT (NFE=2) has comparable sample quality to DDBM (ODE-1, NFE=100) measured by various metrics.
>
> > Q2: It is kind of confusing about the argument with CDDB though. Although the CDDB paper also reports that more NFEs can help, the key is to compare in the same setting?
>
> We apologize for the confusion caused by our argument with CDDB. In this paper, the target of the proposed consistency distillation/tuning is to achieve comparable performance in a few NFEs to their corresponding base model under numerous NFEs. Hence, we mainly compare the performance between the distilled/fine-tuned model (i.e., CBD/CBT) and the corresponding base model (DDBM ODE-1) under different NFEs. We agree that comparing different methods in the same setting is important but we think this should be conducted among different distillation/fine-tuning methods under the same base models targeted for accelerated sampling, which, to the best of our knowledge, is still a vacant area for the diffusion bridge community.
>
> > Q3: The CDDB paper reports that "20 NFE CDDB outperforms 1000 NFE I2SB in PSNR by > 2 db". Can it suppose that CDDB would be a stronger baseline than I2SB to compare with?
>
> We would like to address that CDDB is actually orthogonal to our method since it is a pure inference technique designed for the sampling fashion of diffusion bridges. The sampling process for CBT/CBD can fit into the inference process where CDDB is applicable and thus these two methods can be incorporated together. Moreover, the CDDB requires access to the Gaussian linear measurement for inverse problems, which might be a bit tricky to act as a baseline for fair comparison.

---

### Author Rebuttal · Authors · 2024-08-06

Dear reviewers:

We add some additional results as a part of our response in the one-page pdf, including:
* The demonstration of sample diversity of the deterministic ODE sampler for DDBM (Fig. 9)
* The qualitative comparison between I$^2$SB (their default setup with officially released checkpoint) and CDBM.


We kindly invite you to check the results if they are relevant to your mentioned weaknesses/questions.

Best,
Authors of Submission #17343

---

### Decision · Program_Chairs · 2024-09-25

**Decision:**

Accept (poster)

**Comment:**

This manuscript incorporates ideas from previous work on "consistency models" to generalize recent work on denoising diffusion bridge models, to allow high-quality data generation with a coarser (and thus computationally cheaper) discretization of the underlying ODE.  Reviewers are positive about the approach, but had a number of concerns about technical details, which were ultimately addressed by the author rebuttal.  Please be sure that these clarifications and improvements are incorporated in the final manuscript, and work to make your results accessible to the broader NeurIPS community.